# Van der Waals ferromagnetic Josephson junctions

Linfeng Ai[1,2,9], Enze Zhang[1,9], Jinshan Yang [3,9], Xiaoyi Xie[1,2], Yunkun Yang[1], Zehao Jia[1,2], Yuda Zhang[1,2], Shanshan Liu[1], Zihan Li[1], Pengliang Leng[1,2], Xiangyu Cao[1,2], Xingdan Sun [4], Tongyao Zhang[5], Xufeng Kou[6], Zheng Han[4,5], Faxian Xiu [1,2,7,8 ✉] & Shaoming Dong[3 ✉]

Superconductor-ferromagnet interfaces in two-dimensional heterostructures present a unique opportunity to study the interplay between superconductivity and ferromagnetism. The realization of such nanoscale heterostructures in van der Waals (vdW) crystals remains largely unexplored due to the challenge of making atomically-sharp interfaces from their layered structures. Here, we build a vdW ferromagnetic Josephson junction (JJ) by inserting a few-layer ferromagnetic insulator $Cr_2Ge_2Te_6$ into two layers of superconductor $NbSe_2$. The critical current and corresponding junction resistance exhibit a hysteretic and oscillatory behavior against in-plane magnetic fields, manifesting itself as a strong Josephson coupling state. Also, we observe a central minimum of critical current in some JJ devices as well as a nontrivial phase shift in SQUID structures, evidencing the coexistence of 0 and $\pi$ phase in the junction region. Our study paves the way to exploring sensitive probes of weak magnetism and multifunctional building-blocks for phase-related superconducting circuits using vdW heterostructures.

[1] State Key Laboratory of Surface Physics and Department of Physics, Fudan University, 200433 Shanghai, China. [2] Shanghai Qi Zhi Institute, 41th Floor, AI Tower, No. 701 Yunjin Road, Xuhui District, 200232 Shanghai, China. [3] State Key Laboratory of High Performance Ceramics & Superfine Microstructure, Shanghai Institute of Ceramics, Chinese Academy of Sciences, 200050 Shanghai, China. [4] Shenyang National Laboratory for Materials Science, Institute of Metal Research, Chinese Academy of Sciences, 110016 Shenyang, China. [5] State Key Laboratory of Quantum Optics and Quantum Optics Devices, Institute of Opto-Electronics, Shanxi University, 030006 Taiyuan, China. [6] School of Information Science and Technology, ShanghaiTech University, 201210 Shanghai, China. [7] Institute for Nanoelectronic Devices and Quantum Computing, Fudan University, 200433 Shanghai, China. [8] Shanghai Research Center for Quantum Sciences, 201315 Shanghai, China. [9] These authors contributed equally: Linfeng Ai, Enze Zhang, Jinshan Yang. ✉email: Faxian@fudan.edu.cn; smdong@mail.sic.ac.cn

Recently, two-dimensional (2D) van der Waals (vdW) crystals have been widely investigated on their heterostructures[1,2] by integrating disparate materials, promoting the exploration of unusual phenomena such as charge transfer[3] and proximity effect[4–6]. By taking advantage of excluding the dangling bonds and lattice mismatch at surfaces, high-performance nanodevices have also been demonstrated in electronics[7] and spintronics[8]. Among these, the discovery of intrinsic layered superconductors (S)[9] and ferromagnets (F)[10–12] whose properties vary in layer numbers paves the way to further exploiting such an incompatible order-parameter system. As such, the design of S–F heterostructures mainly focuses on the proximity effect-dominant coupling like Josephson junctions (JJ) with ferromagnetic barriers. With atomically flat and magnetically sharp interfaces, this type of functional nanodevices may potentially serve as critical components for superconducting quantum circuits[13] or memories[14].

In a ferromagnetic JJ, exchange energy ($E_{ex}$) in magnets imposes a finite momentum to Cooper pairs, leading to a modulation of spatially damped oscillations of superconducting order parameter when penetrating the ferromagnetic barrier over the decay length scale[15]. Depending on the JJ barrier thickness, the ground state can either be a 0-junction with equal superconducting phases on both sides or a $\pi$-junction where the phases differ by $\pi$. Consistent with theoretical predictions, experiments have shown the presence of 0-$\pi$ transitions by tuning the F-layer thickness in metallic or alloyed magnets[16,17], as well as 0-$\pi$ JJ structures[18] with a corporate area consisting of these two ground states, providing the potential for an arbitrary $\varphi$-phase control[19] in superconducting electronics. Except for the mentioned above, magnetic insulators in JJs are also fascinating for their spin filtering tunnel behaviors[20], in which a pure second-harmonic current phase relation[21] and macroscopic quantum tunneling[22] have been realized. The nondissipative and low-decoherence nature makes it a promising tool in future applications. Also, inspired by the recent progress in the graphene-based micromagnetometry[23], the JJ technique can be analogically developed to probe the weak magnetism of vdW magnetic insulators, because of its sensitivity to the spin-dependent Josephson tunneling process when applied as a tunnel barrier.

In this work, we report the construction of 2D-vdW JJs with ferromagnetic insulating barriers. Layered dichalcogenide 2H-$NbSe_2$ is an excellent vdW superconductor for its robust superconductivity[24], which has previously been selected to form highly transparent JJs[25,26] and also been known to couple effectively to vdW tunnel barriers[27]. For the other, we choose intrinsic ferromagnetic semiconductor $Cr_2Ge_2Te_6$ with a bulk Curie temperature ($T_{Curie}$) of ~61 K[28] and a narrow gap ~0.38 eV[29] as an intermediate layer, whose easy axis points out of the cleaved plane. The JJ shows a typical Fraunhofer pattern manifesting a strong Josephson coupling state. Besides, the observed hysteresis of critical current ($I_c$) along with the corresponding junction resistance ($R$) against field sweep ascribes to the barrier's magnetism. The measured hysteresis depends on the maximum applied field ($B_{max}$), implying an evolutionary magnetization process of the barrier remanent magnetic moment. Meanwhile, the hysteretic behavior only appears when the JJs are in the superconducting state. We also find a central minimum of $I_c$ patterns in some devices as a typical 0-$\pi$ JJ signature, evidencing the coexistence of both 0 and $\pi$ phase in the whole junction area. Also, a nontrivial ground-state phase $\varphi$ is observed when employing a SQUID structure to detect the phase shift in this system. The possibility of incorporating inhomogeneities in the magnetizations can give rise to the superposition of 0-$\pi$ state via tunneling process, and the asymmetric supercurrent distribution responding to the external magnetic field can be originated from the multi-domain state in the few-layer $Cr_2Ge_2Te_6$.

## Results

**Basic characterizations of vdW JJs.** Ferromagnetic JJ devices are fabricated by a dry-transfer technique[30], which produces a clean interface via vdW coupling between layers. Figure 1a is the top and side view of the layered structures of 2H-$NbSe_2$ and $Cr_2Ge_2Te_6$, respectively. Figure 1b is a schematic illustration of the device structure with a four-terminal measurement configuration, where $NbSe_2$ flakes are placed on the top and bottom and few-layer $Cr_2Ge_2Te_6$ acts as an insulating barrier. The whole stack is fabricated on a silicon substrate capping of 285-nm $SiO_2$. We choose a laminated vertical junction structure that is suitable for a relatively small piece of exfoliated $Cr_2Ge_2Te_6$. Figure 1c displays a false-color scanning electron microscope (SEM) image of device #01 with three layers labeled by the dashed lines, and the height profile of intermediate $Cr_2Ge_2Te_6$ flake is around $6.5 \pm 0.4$ nm measured by the atomic force microscope (AFM), beneath a flat junction area ($S$) of ~14.2 $\mu m^2$. The entire fabrication process is performed in a glove box. A carefully controlled procedure assisted by the stepper motor is applied to eliminate the bubbles and surface oxidation, making it a high-quality vdW interface.

The JJ characteristics are first investigated below the superconducting critical temperature ($T_c$) of the $NbSe_2$ crystals. Figure 1d shows typical current-voltage ($IV$) curves of junction device #01 at 0.1 K under zero field, with a hysteretic behavior in the charge ($I_c$) and retrap ($I_r$) current situations revealing a tunnel junction described by Resistively–Capacitively Shunted-Junction (RCSJ) model, however, the small Stewart-McCumber parameter ($\beta_c$) of ~1.9 is derived from the $I_r/I_c$ ratio of ~0.9 indicating a faintly underdamped regime which seems abnormal for an insulating barrier. The normal state resistance ($R_n$) is 10.6 $\Omega$ obtained from the linear part of its $IV$ curves. In Fig. 1e, a contour plot of junction differential resistance ($dV/dI$) as a function of bias current ($I$) and in-plane field ($B_\parallel$) is presented. Periodic oscillations of $I_c$ from the Fraunhofer pattern correspond well to its in-plane $R$, as the positions of $I_c$ minima are the same as those of $R$ maxima. The periodicity of $I_c$ oscillation $\Delta B$ ~4.2 mT is given by an integer flux quantum $\Phi_0$ in the effective area ($S_{eff}$), from which we can obtain the effective junction length ($d + 2\lambda$) ~82.1–98.5 nm ($d$ is the barrier thickness and $\lambda$ is the London penetration depth of $NbSe_2$ electrodes) with the junction width $W$ ~5–6 $\mu m$ (perpendicular to the field direction), for the area is not so regular. The Josephson penetration depth $\lambda_J = \sqrt{\hbar/2\mu_0 e(d + 2\lambda)J_c}$ is estimated to be 27.4–30.0 $\mu m$ at 0.1 K which is much larger than the junction's lateral size, where $\hbar$ is the reduced Planck's constant, $\mu_0$ is the vacuum permeability, $e$ is the electron charge and $J_c$ (~353.5 A $cm^{-2}$) is the critical current density. Thus our devices should be treated as small JJs. Also, clear second-order oscillations are seen at higher bias due to Fiske resonance[31] when the a.c. Josephson frequency equals the eigenfrequency of electromagnetic oscillations in this cavity, proving a well-defined JJ through the vdW stacking. Returning to the cavity nature of our JJ, the specific capacitance $C_s$ can be also derived from the first voltage position $V_1$ (0.57 mV) of the Fiske steps and the plasma frequency $\omega_p$, calculated as 22.3–26.9 fF $\mu m^{-2}$ which has the similar magnitude as obtained for $Cr_2Ge_2Te_6$[32] ($C_s$ ~5.5–21.8 fF $\mu m^{-2}$, the dielectric constant $\varepsilon$ is ~4–16). And the parameter $\beta_c$ is given by the relationship $\beta_c = 2eI_cR_n^2C/\hbar$ as 5.4–6.7, larger than the estimation using the RCSJ model due to thermal activation or noises in the external circuits of $IV$ measurements[33], making it complicated to evaluate the internal dissipation.

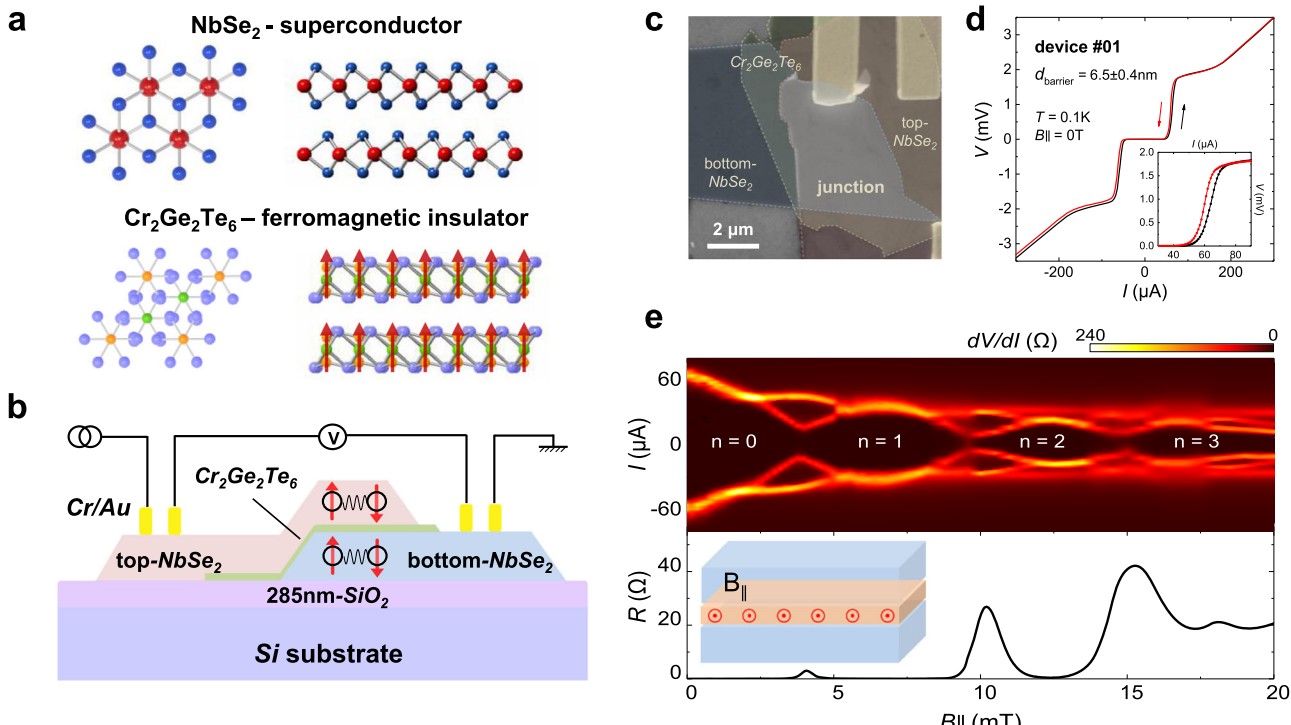

**Fig. 1 Device structure and basic characterizations of vdW Josephson junctions based on NbSe₂ and Cr₂Ge₂Te₆ flakes.** **a** Crystal structures of superconductor 2H-NbSe₂ (Nb, red and Se, blue) and ferromagnetic insulator Cr₂Ge₂Te₆ (Cr, green, Ge, orange and Te, purple). The top and side views show their layered structures. Red arrows in Cr₂Ge₂Te₆ indicate the magnetization. **b** Schematic illustration of the junction with the measurement configuration. A four-terminal measurement on the laminated vertical structure is performed. **c** False-color scanning electron microscope (SEM) image of device #01 (scale bar: 2 μm). The three layers are labeled by the dash lines, respectively, and the junction area is ~14.2 μm². **d** Typical current-voltage $IV$ curve of JJ (device #01) at 0.1 K under zero magnetic field with the charge (black) and retrap (red) situation. The thickness of the barrier is ~ 6.5 nm measured by atomic force microscope (AFM). The inset is a zoom-in plot of its $IV$ curves. **e** Differential resistance d$V$/d$I$ map as a function of bias current $I$ and in-plane magnetic field with a periodically oscillatory junction resistance, showing a Fraunhofer pattern. Besides, higher-order oscillations due to Fiske resonance indicate a highly transparent junction surface.

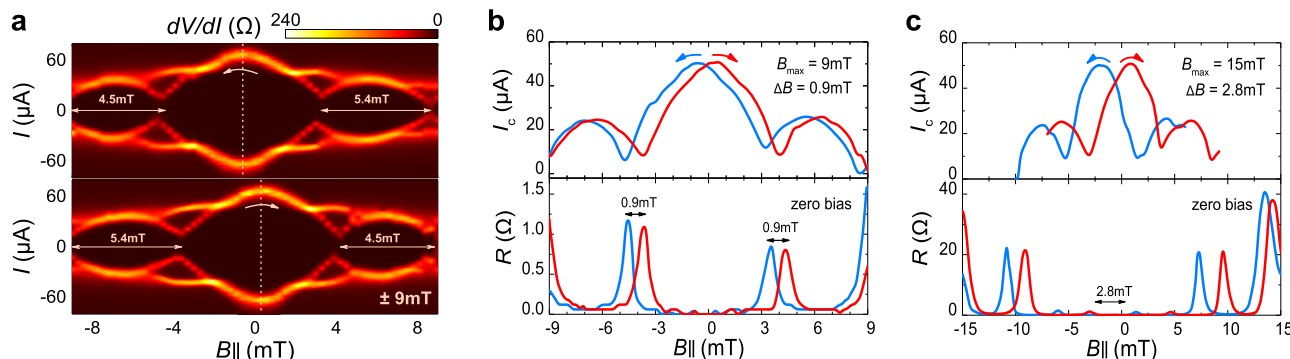

**Fig. 2 Hysteretic Fraunhofer patterns of the unsaturated barrier under in-plane magnetic field.** **a** Critical current $I_c$ oscillation patterns of device #01 when the magnetic field sweeps from ±9 mT (upper panel: positive to negative; lower panel: inverse). The yellow dashed lines mark out the central maximal of $I_c$, with respect to distinct field positions. **b** Hysteresis of $I_c$ and the corresponding junction resistance $R$ starting from ±9 mT. The effective field seen by the junction has an additional value of 0.9 mT originating from the remnant barrier moment. **c** Similar hysteresis loop of $I_c$ and $R$ for the maximum applied field $B_{max}$ of ±15 mT, which gives a larger contribution to the total flux in the junction of 2.8 mT for a magnetic barrier whose magnetization is still unsaturated.

**Evolution of the magnetic barrier in JJs.** For an unsaturated magnetic barrier, it is feasible to continuously reverse its orientations at a lower magnetic field. In Fig. 2, we present a comparison of Fraunhofer oscillation patterns of device #01 (~6.5 nm) when the field starts from different $B_{max}$. Initially, it was magnetized to 9 mT at 0.1 K, then descended from positive to negative fields at −9 mT and eventually returned. The separated sweep branches in Fig. 2a give the central maximal $I_c$ offsets at −0.6 mT

and 0.3 mT, respectively. The total flux enclosed in the junction is the sum of the external flux and additional flux generated by the barrier magnetic moment, the latter of which reverses smoothly at a low magnetic field, resulting in a hysteresis window of 0.9 mT. The shape of patterns is slightly distorted under the magnetized process, as the field widths of the first side-lobes of the pattern are not exactly the same as those at opposite field directions, consistent with the model of a field-dependent magnetic moment in a

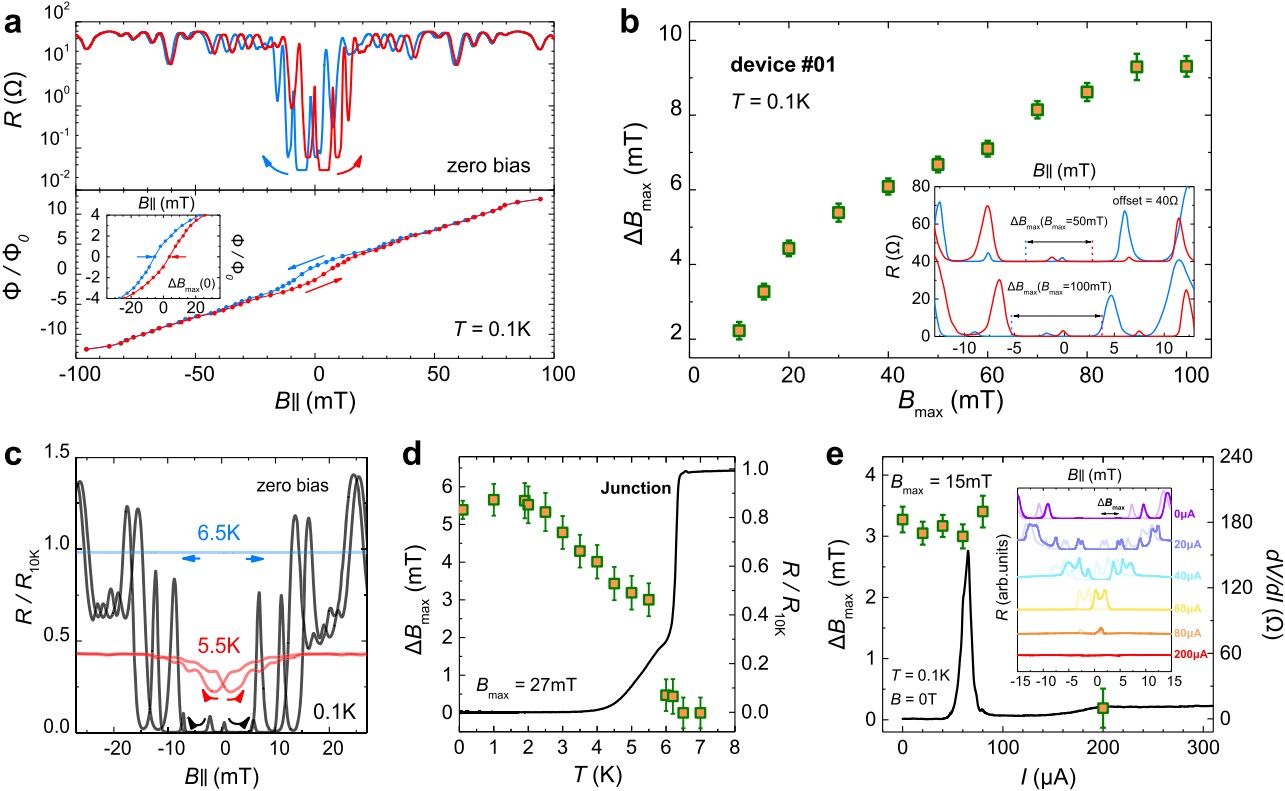

**Fig. 3 Evolution of the barrier magnetization in the Josephson junctions. a** (upper) Hysteretic in-plane $R$ (log-plot) and (lower) the field dependence of trapped flux in the junction of device #01 with $B_{max}$ of 100 mT at 0.1 K. The latter depicts the magnetization curve of the barrier itself. The largest hysteresis $\Delta B_{max}$ estimated from the central position of its zero-order minimum of $R$ is 9.3 mT. The inset is a zoom-in plot near the zero magnetic field. **b** Dependence of $\Delta B_{max}$ about $B_{max}$. It performs a continuous increase and finally converges at higher fields. The inset is the comparison of hysteretic $R$ and $\Delta B_{max}$ with a $B_{max}$ of 50 and 100 mT (offset of the former is 40 Ω for clarity). Error bars indicate the standard error in readouts of the hysteresis by resolutions of data points, and below are the same. **c** In-plane $R$ compared between the superconducting state and normal state. Magnetic hysteresis shrinks with temperature increasing, then disappears above $T_c$ of NbSe₂, suggesting that the magnitude of superconducting interference with temperature reflects in the size of hysteresis. **d** Temperature dependence of normalized junction $R$ and $\Delta B_{max}$ ($B_{max}$ = 27 mT). Value of the latter one remains almost unchanged below 2 K and gradually decreases up to 6.4 K. **e** $\Delta B_{max}$ ($B_{max}$ = 15 mT) at different d.c. bias current along with zero field d$V$/d$I$ curve at 0.1 K. $\Delta B_{max}$ is identical (within range of error) below 100 μA but virtually zero at high bias, indicating the disappearance of hysteresis. The inset is the hysteretic $R$ at 0–200 μA bias current from superconducting to the normal state (stacked and colored for clarity).

hysteresis loop. The hysteresis observed in zero-bias $R$ is equivalent to $I_c$ (Fig. 2b). Next, we magnetized the device to 15 mT and repeated the same measurement at the higher applied field. Figure 2c shows that the positions of zero-flux $I_c$ shift to −2 mT and 0.8 mT, resulting in a larger window of 2.8 mT. Estimations from $I_c$ and $R$ measurements both give accordant sizes of hysteresis.

To further understand the evolution of barrier magnetization ($M$), first, we obtain its hysteretic in-plane $R$ (log-plot is used to clarify zero $R$) with a $B_{max}$ of 100 mT at 0.1 K (see the upper panel in Fig. 3a), in which the flux-induced oscillations in $R$ persist when the supercurrent is unmeasurable at higher fields. The coincidence over ±75 mT of these two sweep directions reveals a saturation field of its moment. Empirically, we can correspond each minimum/maximum of $R$ to half-integer/integer flux quantum trapped in the junction[34] (see the lower panel in Fig. 3a). Therefore, the $M$ of the barrier can be reconstructed from the equation $\Phi = 4\pi MWd_F + \Phi_H$ ($d_F$ is the thickness of the ferromagnetic interlayer), where $\Phi$ is the sum-up flux embedded in a JJ and $\Phi_H$ is generated by the external field. Meanwhile, the largest hysteresis ($\Delta B_{max}$), estimated from the central position of zero-order minimum $R$, can also be regarded as a quantitative index for this nanoflake's in-plane magnetization (in device #01, $\Delta B_{max}$ is 9.3 mT when fully saturated to 100 mT). Next, we show

the plot of $\Delta B_{max}$ varying with $B_{max}$ at 0.1 K in Fig. 3b, from which we find that $\Delta B_{max}$ is proportional to $B_{max}$ and nearly converges above 90 mT.

We also investigate the hysteretic behavior with thermal excitation, conjointly resolving the features about how hysteresis vanishes above the superconducting $T_c$. Figure 3c depicts the hysteretic in-plane $R$ of the same device compared between the superconducting state and normal state. The hysteresis shrinks in $R$ amplitude and magnetic field range as temperature increases and finally disappears above $T_c$ since it is non-superconducting, which seems contradictory to the fact that Cr₂Ge₂Te₆ at 6.5 K is still in the ferromagnetic state, as the hysteresis originates from its remanence. For detailed analyses, we measure the variation of $\Delta B_{max}$ as a function of temperature with a $B_{max}$ of 27 mT along with the normalized $R$ under zero field (Fig. 3d). The value of $\Delta B_{max}$ remains almost unchanged below 2 K and then gradually decreases when warming up to 5.5 K. Nevertheless, the discrepancies in the curves of 6 and 6.2 K turn to become illegible around zero field causing a sudden drop of $\Delta B_{max}$ (see Supplementary Fig. 5). Therefore, we assume that $\Delta B_{max}$ here can denote the magnitude of Josephson interference. The reduction of $\Delta B_{max}$ is predominantly determined by thermal fluctuation above 2 K, where the phase coherence between the two separated superconductors is suppressed at high temperatures near $T_c$. This assumption is further

supported by the bias-dependent experiments. A set of hysteretic $R$ loops at *d.c.* bias current varying from 0 to 200 μA is shown in Fig. 3e, from which $\Delta B_{max}$ is obtained when the junction is driven into several superconducting states (qualitatively distinguished in the 0 mT $dV/dI$ curve). We can observe an identical $\Delta B_{max}$ around 3.3 mT (within the error bar) below 100 μA, in line with our expectations that $\Delta B_{max}$ is only influenced by a specific $B_{max}$ before saturation at a specific temperature. Besides, the deviation between two directional sweeps is practically unidentifiable at a high bias of 200 μA and 400 μA (see Supplementary Fig. 6) in good agreement with a relatively small $\Delta B_{max}$ measured above 6 K, as a consequence of the main contribution to the junction $R$ from the normal state $R$ of superconductor $NbSe_2$.

**Evidence of π phase coupling.** For those spin-singlet pairing superconductors, the presence of a π-JJ owing to the orbital effect in a metallic barrier requires direct exchange interactions and a comparable F-layer thickness to the oscillation wavelength. The Josephson phase is correlated through the whole junction. As the spontaneous flux penetrations into 0 and π phase parts produce a counterproductive effect, they will conjointly result in a central minimum of $I_c$ instead of a maximum value either in absolute 0-JJs or π-JJs. Nevertheless, the realization of a π-JJ in the tunneling regime with the use of ferromagnetic insulators has been proposed theoretically on the atomic scale model[35], but still waited for more explicit explorations. Here, we first find similar signatures of 0-π JJs in our devices which may also be the evidence of a φ-phase state. Figure 4a is the consequent non-Fraunhofer patterns of device #02 (the area $S \sim 19.6\ \mu m^2$) in the range of ±30 mT, whose barrier thickness is around 10.5 ± 0.3 nm. Extracted $I_c$ values at a positive bias ($I_c^+$) are given in Fig. 4b and the hysteresis with regard to the opposite sweeping direction is around 3 mT, acquired from the valley positions of the central minimum of $I_c$. Besides, the two sweeping branch curves of $I_c$ coincide above 30 mT, implying a smaller in-plane magnetic

remanence of a thicker $Cr_2Ge_2Te_6$ than that of a thinner one measured in device #01.

The birth of a negative supercurrent in the insulator JJs is worthy of discussion. In analogy to the resonant tunneling process of Cooper pairs coupled by a single-level interacting quantum dot[36], the spin-order of the Cooper pair transferring through a single electron with a given spin orientation will be reversed to conserve the momentum, resulting in a significant change of the single pairing state to a π-shift through fourth-order co-tunneling process (Fig. 4c). When extended to a tunnel barrier with inhomogeneous magnetizations, if the momentum of a Cooper pair is parallel to the same direction as the spin orientation in $Cr_2Ge_2Te_6$, a segment of π phase coupling tunneling, therefore, inhibits the positive supercurrent from the normal 0 phase. As illustrated in Fig. 4d, junction region with precisely perpendicular magnetizations will result in π-JJs while other parts whose magnetizations are tilted away will form the rest 0-JJs. Another feasibility is to consider the magnetic activity of the barrier assuming nonuniform exchange interactions, inspired by the recent evidence of an incomplete 0-π transition in thick spin-filter JJs[37]. The spin-mixing angles for different transport channels are close to π as the barrier thickness increases exceeding the superconductor's coherence length perpendicular to the layers ($\xi_0 \sim 2$–3 nm[38]), which also implies a spin-triplet correlation. Besides, compared with the long junction of a stepped F-layer barrier, we didn't observe a tiered flake of $Cr_2Ge_2Te_6$ with distinguished two regions of different thickness in our JJs, whereas the nonconventional Fraunhofer patterns could be introduced by a magnetization texture of 0-π stacks with inhomogeneous exchange field[39].

To further detect the phase shift in our systems, we also take phase-sensitive measurements employing a SQUID structure[40] consisting of an SFS JJ ($NbSe_2/Cr_2Ge_2Te_6$) and a reference JJ ($NbSe_2/NbSe_2$). Details of the fabrication process are given in "Methods". Figure 5a displays the false-color SEM image of

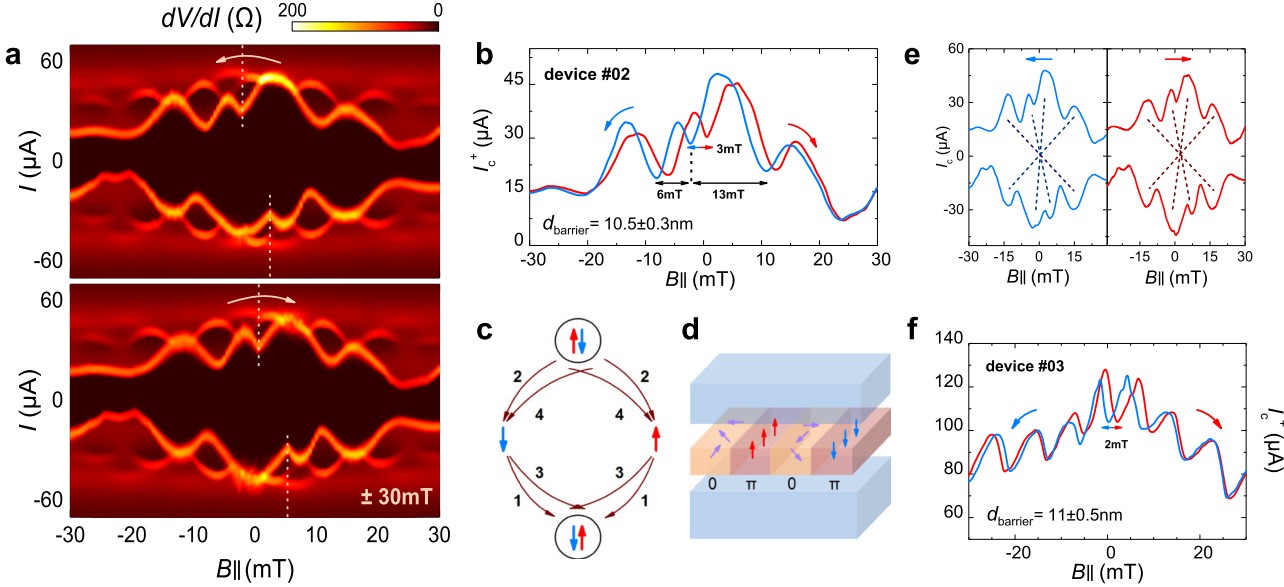

**Fig. 4 Unconventional Fraunhofer patterns as evidence of π phase coupling. a** 0-π junction like $I_c$ oscillation patterns of device #02 with the magnetic field starts from ±30 mT at 0.1 K. Its barrier thickness is ~10.5 nm. The yellow dashed lines mark the valley positions of the central minimum of $I_c$ to depict a 0-π junction behavior. **b** Hysteresis of $I_c^+$ of device #02. The $\Delta B_{max}$ value is 3 mT estimated from the valley positions in the center. The difference of field widths around this valley indicates an asymmetric local magnetization of the barrier. **c** Origin of a π phase through the magnetic tunneling of Cooper pairs. The reversal of the spin-order involves the fourth-order co-tunneling process. **d** Illustration of a junction area with several π segments (perpendicular magnetizations) and 0 segments (tilted or parallel magnetizations). **e** Comparisons of $I_c$ at the positive and negative bias of device #02 (left and right: opposite sweeping branches) to exclude the influence of trapped vortex. **f** Another 0-π junction signature of device #03 with a ~11-nm-thick barrier. The field starts from ±50 mT at 0.1 K, giving the $\Delta B_{max}$ value of 2 mT.

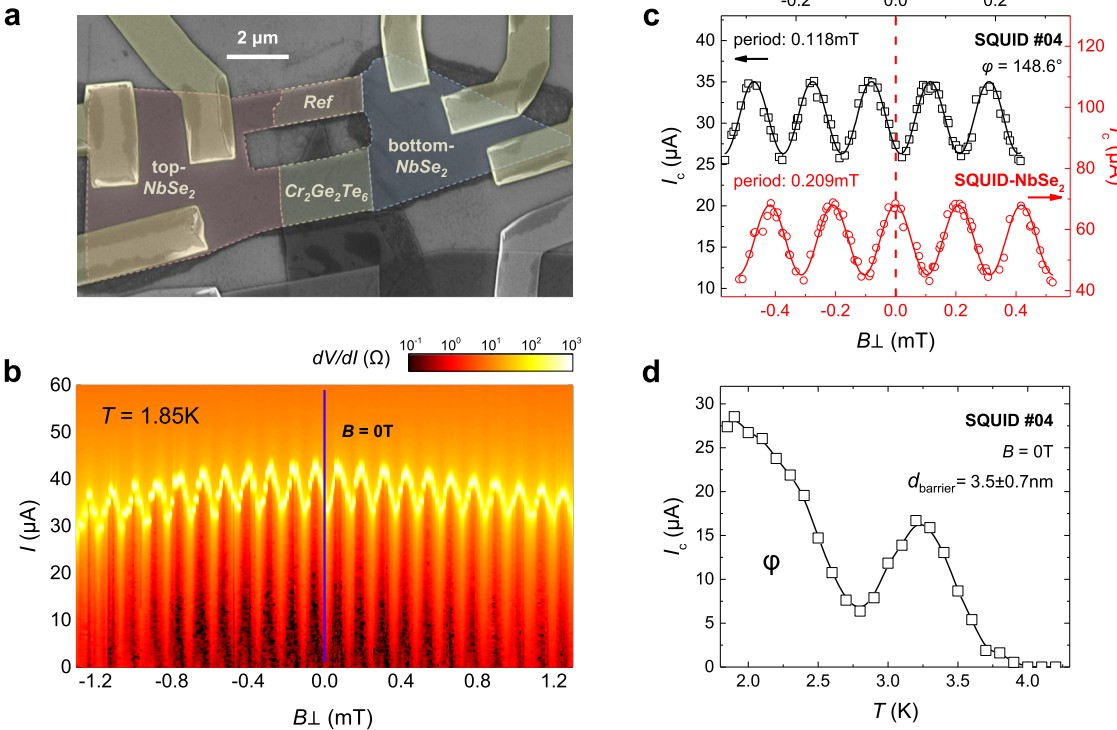

**Fig. 5 Josephson phase measurements. a** False-color SEM image of SQUID #04 (scale bar: 2 μm), consisting of an SFS JJ with a $Cr_2Ge_2Te_6$ barrier and an intrinsic reference JJ formed by two $NbSe_2$. The area of the SQUID loop (17.5 μm$^2$) is the sum of the empty parts (~5.9 μm$^2$) and the screening areas (~11.6 μm$^2$). **b** Differential resistance d$V$/d$I$ (log-scale) map of SQUID #04 as a function of bias current $I$ and out-of-plane magnetic field measured at 1.85 K, showing a SQUID oscillation. All the d$V$/d$I$ curves are sweeping from zero to positive bias. The position of zero magnetic field calibrated by a $NbSe_2$ SQUID on the same substrate chip is indicated by the purple line in this plot. **c** Critical current $I_c$ oscillations of SQUID #04 (black rectangles) with the period of 0.118 mT, consistent with the SQUID loop's area. The phase φ of it is about 148.6°, obtained by a fit to $I_c$ with the sine relation (black lines). The red circles show the $I_c$ oscillations of the $NbSe_2$ SQUID for calibration with the period of 0.209 mT and the red lines are the fitted results. **d** Temperature dependence of $I_c$ of SQUID #04 at zero magnetic field. The thickness of the $Cr_2Ge_2Te$ barrier is ~3.5 nm measured by AFM. The nonmonotonic change of $I_c$ implies an incomplete 0-π transition in SQUID #04 with a φ ground-state phase at low temperature.

SQUID #04, and its transport results as SQUID oscillations in $I_c$ measured at 1.85 K are presented in Fig. 5b. The SQUID $I_c$ as a function of magnetic flux $Φ$ is counted through the SQUID loop (17.5 μm$^2$) as the sum of the empty areas (5.9 μm$^2$) and screening areas (11.6 μm$^2$), agreed with the area calculated from the oscillation period of 0.118 mT, as shown in Fig. 5c. To extract the phase of the SQUID #04, we have calibrated the measurement systems through a $NbSe_2$ SQUID on the same substrate to define the zero magnetic field, the data of which is also plotted in Fig. 5c. Under this condition we can find that the position of maximum $I_c$ in the central period of SQUID #04 is deviated from the zero magnetic field, providing one nontrivial phase φ = 148.6° rather than a 0 or π phase by a fit with sine relation. It is known that a JJ with an arbitrary value φ of the phase should possess doubly degenerate ground states ±φ, however, the high damping environment in our JJs prohibits us to observe the other phase and its corresponding lower $I_c$ following the specific bias sweeping sequences[19].

In addition to the phase shift addressed in SQUID #04, we then measure its $I_c$ in the variation of temperature and surprisingly find a nonmonotonic dependence which is likely to be an induced 0-π transition occurring at $T$ ~0.8 $T_c$ in this device, as shown in Fig. 5d. The thickness of the barrier is about 3.5 ± 0.7 nm. The shape of the $I_c(T)$ curve is a more standard cusp-like 0-π transition with a higher weight of the s-wave singlet component, corresponding to the relatively low values of disorder and spin-mixing effects in this thin barrier. Compared to the similar phenomenon in GdN-based spin-filter JJs introduced above, the interfacial inhomogeneities, the high spin-filter efficiency of the barrier, and a large spin-mixing angle will result in the stronger magnetic activities in these magnetic JJs, in which the deviations of $I_c$ from the conventional Ambegaokar–Baratoff (AB) behavior[41] are more evident for thicker barrier samples. More investigations on the regime adapted for full-range thickness are needed to explore how the ground-state phase evolutes and the realistic micromagnetic nature of barriers could be taken into account.

## Discussion

Our experiments on ferromagnetic JJs via vdW connection demonstrate a hysteretic supercurrent and resistance behavior, dominated by the barrier magnetization related to the $B_{max}$ and temperature. Since the unsaturated barrier moment can reverse smoothly at low fields, the supercurrent here follows a continuous modulation by the external flux shown in $I_c$ measurements above. However, with progressive magnetization to a higher field, the Josephson tunneling process tends to become fragile, as the emergence of breakpoints in $I_c$ causes a discontinuous pattern (see Supplementary Figs. 7 and 8). In device #01, the discontinuity starts from 23 mT, with an evident signature of abrupt changes of $I_c$ between two flanks of a breakpoint. Another feature is the irreversibility of the whole pattern as the breakpoints persist when the field sweeps back. Detailed characterizations are performed in the Supplementary Information. This phenomenon

might be associated with the layered spin structures of $Cr_2Ge_2Te_6$, accounting for its enhanced interlayer coupling at high fields, that the barrier magnetization will resist external changes, thus a larger step of the applied field is required to overcome such a potential. When the fully-aligned spins start to flip back, the spin interactions between separated layers prevent it from a smooth reverse leading to the observed irreversibility. Besides, pinned spins of the ferromagnetic layer adjacent to the superconducting layer may also be restricted due to their 2D circumstances, strongly modulated by impurities and obstacles at two heterointerfaces.

Referring to the shape of unconventional Fraunhofer patterns, it should be mentioned that the influence of Abrikosov vortex[42] from superconductor electrodes trapped in the JJ can introduce anomalous phase shifts as well, leading to similar observations. There are two criteria to distinguish that, one is the vortex-induced hysteresis that is opposite to remanent magnetization[43,44], which is contradictory to our observations. The other can be inferred from the symmetry of its interference pattern. For a conventional JJ, $I_c$ should be symmetric with respect to $I_c^+(B) = I_c^-(B)$ and $I_c(+B) = I_c(-B)$. Distorted $I_c$ patterns by the vortex will naturally accord with the former formula, yet neither of them has been satisfied in our devices' results. The crossing dashed lines in Fig. 4e reveal that the symmetry of our JJs could only be preserved when both considering the current and magnetic field inversion operations. Furthermore, deviated from a symmetric 0-π junction model, we find several remarkable asymmetries of the $I_c$ patterns in our data. First, the nonzero central minimum of $I_c$ can be originated from the asymmetry in 0 and π phase critical current densities. Besides, unequal $I_c$ values and lobe widths of the two halves of subsequent peaks around the central valley should ascribe to different field-dependent flux penetration to 0 and π parts, introduced by the barrier in-plane magnetization[18]. Last, the nonequivalent periodic dependence of $I_c$ oscillations compared between the positive and negative field applied halves (6 and 13 mT, marked in Fig. 4b) indicates a strong modulation by inhomogeneous local magnetizations. The nature of a possible multi-domain state in $Cr_2Ge_2Te_6$[45] with randomly distributed magnetizations is expected to generate such asymmetries in this ferromagnetic barrier. The pinning of the domains can be reconfigured under the external in-plane magnetic field in the two halves, therein resulting in the observed variations of the central minimum as well as a successive maximum of $I_c$ values with regard to the opposite field sweeping directions. Similar behaviors also perform in another device #03 ($S \sim 23.5 \mu m^2$) (Fig. 4f) whose barrier thickness is of the same magnitude of around $11 \pm 0.5$ nm, giving the hysteresis of around 2 mT with a $B_{max}$ of 50 mT. Both two devices in this part have a comparative $I_c$ background due to the finite-voltage criterion effect.

Also, we want to discuss the comparisons of JJ characteristics varying in barrier thickness. For a thinner barrier, the temperature dependence of $I_cR_n$ product ($V_c$) of device #01 ($\sim 6.5$ nm) is a well fit to AB relation for an insulating barrier. The calculated zero temperature $V_{c0} \sim 0.51$ mV is much smaller than the theoretical maximum value of $\pi\Delta/2e = 1.57$ mV ($\Delta \sim 1$ meV is the gap value of bulk NbSe2), showing a strong suppression by the magnetic barrier on Josephson current for the phase differences acquired by spin-up or spin-down electrons. Compared to the suppression, we find an apparent excess supercurrent over the AB relation fit at low temperature (see Supplementary Fig. 9) in the thicker Josephson device #03 ($\sim 11$ nm). The $I_cR_n$ product measured at 0.5 K exceeds the calculated limit from the AB relation fit. The gradual deviation between the data and AB values below 4.6 K is similar to former studies in NbN/GdN/NbN JJs[20]. Nevertheless, we didn't observe any evidence of 0-π transition in its temperature dependence. Considering our material systems,

the equilibrium domain size of $Cr_2Ge_2Te_6$ is probably larger in ultrathin nanoflakes, while the formation of multidomains is observed in the thicker sample with a smaller domain size. Further discussions on the correlation between the domain size and the film thickness are presented in Supplementary Information. Aware of the discrepancy in barrier thickness, the relative size of domain walls would introduce a notable magnetic inhomogeneity that implies the reduction of exchange energy seen by the Cooper pairs, therein promoting the enhancement of Josephson current in thicker samples[46].

We would refocus on the validity of using the JJ structure to probe magnetism in atomic vdW materials. Few experiments have shown a direct study of ferromagnetic semiconductors or insulators via transport measurements, particularly limited at low temperatures. Like previously reported structures of tunneling magnetoresistance (TMR)[47] or spin–orbit torque (SOT)[48] devices for the same goal, the JJ is also a potential structure for its high sensitivity to minor variations in the response of external magnetic fields, because of a dramatic transition between the normal and superconducting states. The thought of employing flux quantization as an in-situ magnetometer proves it to be a precise technique to trace the evolution process of its barrier magnetization.

To conclude, we report the realizations of Josephson junctions based on 2D van der Waals superconductor NbSe2 and ferromagnet $Cr_2Ge_2Te_6$. We demonstrate that such hysteretic behavior of supercurrent and magnetoresistance is predominantly induced by the magnetic barrier remanence, therefore the hysteresis can be modulated by the maximum applied field and temperature. Moreover, we observe 0-π JJs signatures which are the evidence of π phase coupling in this crystalline vdW heterostructure, possibly attributed to a momentum-conserving tunneling process through the region with multidirectional magnetizations. And the layered structure of the barrier can generate an anomalous tunneling effect for its internal interlayer coupling. The stacked nanojunctions are of great potential to investigate vdW coupling in layers, and further expectations on dissipationless spin-active or switchable φ-phase JJs can be accomplished in the use of other insulating magnets like $CrI_3$ or $CrCl_3$ with intrinsic anisotropic magnetization.

## Methods

**Crystal growth**. A high-quality crystal of bulk NbSe2 was grown via the chemical vapor transport method with iodine as the transport agent. A stoichiometric ratio of Nb and Se powders (with 0.2% excess of Se) were evacuated and sealed in a quartz tube with 0.1 g iodine flakes and then placed in a two-zone furnace in a temperature gradient from 730 °C to 770 °C for 2 weeks. Single-crystals of $Cr_2Ge_2Te_6$ were synthesized using the Te self-flux method. The raw powders of Cr, Ge, and Te (with a ratio of 1:4:20) were mixed and kept at 950 °C for 6 h, and the mixture was then cooled at a rate of 2 °C/h, followed by centrifugation at 500 °C to get its bulk material.

**Device fabrication**. The heterostructure devices were fabricated through a dry-transfer method in the glove box (H2O, O2 <0.1 ppm). 2H-NbSe2 and $Cr_2Ge_2Te_6$ flakes were first mechanically exfoliated onto polydimethylsiloxane (PDMS) stamps with rough identifications of thickness by optical microscope, and next stacked onto a Si substrate capped with a 285-nm-thick SiO2 layer slowly, controlled by a stepper motor. A piece of flat hBN could be placed on the top to cover the whole stack to protect the sample from further oxidation. The multiterminal electrical contacts were patterned by a standard e-beam lithography technique and subsequently deposited of Cr/Au (5/70 nm) via magnetron sputtering. The process for a SQUID is a little different. During the assembly, the middle $Cr_2Ge_2Te_6$ layer was partially covered along one side of the bottom NbSe2 flake, then the top NbSe2 was placed onto these two parts with $Cr_2Ge_2Te_6$ in beneath or not. To define the SQUID structure, we used e-beam lithography to prepare the patterned masks. Redundant parts were finally removed by reactive ion etching with a mixture of SF6 and O2.

**Transport measurements**. The transport measurements were carried out in two Physical Property Measurement Systems (PPMS, Quantum Design), one equipped with a dilution refrigerator with the temperature down to 50 mK and the other without down to 1.85 K. The transport properties were acquired using lock-in

amplifiers (SR 830) and Agilent 2912 meters. In differential resistance measurements, a small *a.c.* excitation was generated from the lock-in amplifier output voltage in combination with a larger *d.c.* bias applied by Agilent 2912 through a $100 \, k\Omega$ resistor, and the differential voltage was measured at a low frequency (<40 Hz). In magneto-transport measurements, the field direction was parallel to the substrate, therefore perpendicular to the current flowing across the barrier from top to bottom superconductors.

**Analysis of the JJ capacitance from Fiske steps**. To get the capacitance of the JJ[31], first, the voltage positions $V_n$ of Fiske resonance is given as $V_n = n\Phi_0 c/2W$, where $c$ is Swihart velocity and $W$ is the junction width. Using the Josephson penetration depth $\lambda_J$ determined from the Fraunhofer pattern, the plasma frequency of a JJ is obtained as $\omega_p/2\pi = c/\lambda_J$. Moreover, the plasma frequency is defined as $\omega_p = 1/\tau_p = [2eJ_c/\hbar C_s]^{1/2}$ where $\tau_p$ is the time constant, therefore we could acquire the specific capacitance $C_s$.

**Ambegaokar–Baratoff theory**. Also, to research the temperature dependence of our JJs, we used the AB theory[41] to fit the data, expressed as $I_c(T)R_n = (\pi\Delta(T)/2e)\cdot\tanh[\Delta(T)/2k_BT]$, where $\Delta(T)$ is the superconducting gap calculated from BCS theory and $k_B$ is the Boltzmann constant.

## Data availability
The data that support the plots within this paper and other findings of this study are available from the corresponding author upon reasonable request.

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

## Acknowledgements
This work was supported by the National Natural Science Foundation of China (51772310, 11934005, and 11874116), the National Key Research and Development Program of China (Grant No. 2017YFA0303302 and 2018YFA0305601), the Science and Technology Commission of Shanghai (Grant No. 19511120500), the Shanghai Municipal Science and Technology Major Project (Grant No. 2019SHZDZX01), and the Program of Shanghai Academic/Technology Research Leader (Grant No. 20XD1400200). E.Z. acknowledges support from China Postdoctoral Innovative Talents Support Program (Grant No. BX20190085) and China Postdoctoral Science Foundation (Grant No. 2019M661331). Part of the sample fabrication was performed at Fudan Nano-fabrication Laboratory. We sincerely thank Dr. Ce Huang from Fudan University for tremendous help to all results. We thank Xuejian Gao from Prof. Kam Tuen Law's group

from Hong Kong University of Science and Technology for helpful discussion on the Josephson interference mechanism.

## Author contributions

F.X. conceived the ideas and supervised the overall research. Z.J. and Y.Z. synthesized high-quality NbSe$_2$ bulk samples. X.S. synthesized high-quality Cr$_2$Ge$_2$Te$_6$ bulk samples. L.A., Z.J., and Y.Z. fabricated the nanodevices. L.A. and E.Z. performed the PPMS measurements and data analysis. X.X., Y.Y., and X.C. provided the curve fitting. S.L., Z.L., and P.L. carried out the SEM, AFM, and SQUID measurements. T.Z. performed the Kerr rotation measurements. X.K. and Z.H. provided in-depth discussions of vdW physics. J.Y. and S.D. provided high-quality boron nitride samples and helped with some material characterization. L.A. and F.X. wrote the paper with assistance from all other co-authors.

## Competing interests

The authors declare no competing interests.
