## [Peer Review File · Nature Communications]

REVIEWER COMMENTS

Reviewer #1 (Remarks to the Author):

In this manuscript, Ai and co-authors fabricated Josephson junction built with vdW interfaces using NbSe₂ as a superconductor and Cr₂Ge₂Te₆(CGT) as a ferromagnetic insulator. The author demonstrates the manipulation of Josephson critical current through the magnetism of CGT. The superconductor/ferromagnet/superconductor junction have received lots of interest, yet experiments were limited in the difficulty of fabrication of high quality devices using conventional deposition techniques. Here, Ai et al. show a possibility to fabricate high quality Josephson junction (that is confirmed by the observation of Fiske resonance) and observed cooperative phenomena between supercurrent and magnetism. The subject of this study can contain broad interest for solid state physics, therefore could be suitable for the aim of the submitted journal.

Although the results seem to be interesting, I found the discussion and interpretation of the results are not clear. There are quite a few points that discussions are rather hypothetical. I believe these points need to be significantly improved for the publication in the submitted journal. Please consider the following suggestions.

(1) In the analysis, author said $(d+2*\lambda)$ is ~ 82 to 98 nm with $d \sim$ few nm and London penetration depth, λ is few-tens of nm. The London penetration depth exceeds their NbSe₂ thickness (NbSe₂ is a few layers thick according to their method section). It is not clear whether they can use this analysis when the penetration depth exceeds the thickness of NbSe₂. Or did the author actually use a thickness of few-layer NbSe₂ instead of λ ? Please add some explanation for this.

(2) I am concerned about the influence of the fringing field generated from the CGT. Since CGT can become a multi domain structure, there is a fringing from the domain wall. Please explain how authors can distinguish between the contribution of the fringing field and the interpretation they provided in this manuscript.

(3) The following argument highly depends on the magnetic anisotropy constant and probably exchange stiffness and some other parameters of the ferromagnet. Authors need to justify this discussion by looking at literature or additional experiments.

"Considering our material systems, the equilibrium domain size of Cr₂Ge₂Te₆ is probably larger in ultrathin nanoflakes as mediated by a single-domain remanence, in contrast to the tendency of forming multi-domains when approaching a 3D bulk limit. "

In addition to this, the magnetism of their CGT flake is not clear at all. Please add more detail characterization on the magnetism of the CGT material.

(4) The magnetic field-induced hysteresis author discussed in Fig.2 and Fig.3 is a rather small value and definition of exact zero magnetic field can be difficult. How did the author make sure that this is not the artifact of measurement instruments such as the remnant field of the coil?

(5) In Fig. 3(d) and 3(e), the author demonstrates the temperature and current dependence of the results. Is it possible that the superconductivity (or Josephson junction) is sensitive to probe the phenomena that makes observables below T_c of NbSe₂, yet phenomena itself still remain above T_c but there isn't enough sensitivity to detect it above the T_c .

(6) The section named "Evidences of π (π) phase coupling" does not contain a clear explanation to justify their interpretation and need to be improved. I am not sure that the presence of π (π) phase coupling is justified in the experiment. For example, the asymmetry in the current can be possible with Oersted field contribution on the Josephson junction measurement (please see page 57 of http://www.wmi.badw.de/teaching/Lecturenotes/AS/AS2013_Chapter2_Slides.pdf). Maybe the author requires phase sensitive measurement on the Josephson critical current or makes correlation between the real magnetic structure of their CGT in the junction and transport data by some means.

(7) I was assuming that the author uses a few-layer thick NbSe₂ throughout their manuscript according to their method section. However there is a sentence "bi-lateral few-layer NbSe₂ (20 ~ 40 nm)" in supplementary information, which seems to be confusing. Please confirm all the thickness of NbSe₂ and specify in each device.

(8) Since they have data for CGT thickness dependence of junction's temperature dependence, I think

they can also check the CGT thickness dependence of junction RA(resistance area product). Please include this information which may further support the change of transport property change with the thickness of CGT.

Reviewer #2 (Remarks to the Author):

In the manuscript of L. Ai et al, a van der Waals type of magnetic Josephson junction is constructed by stacking superconducting NbSe₂ and ferromagnetic insulator CGT. The transparent interface allows for the observation of a complete Fraunhofer pattern and the associated Fiske resonance. Detailed characterization of hysteresis are conducted as a function of in-plane magnetic fields, bias current and temperatures. In particular, for a thicker ferromagnetic layer, the JJ exhibits a dip of critical current around zero Tesla, which is claimed to be a signature of ϕ -JJ.

The devices seems to be of high quality and the measurement results are in details. Nevertheless, there are several issues to be clarified.

1. In Fig. 2-4, a striking hysteresis is observed upon sweeping the in-plane B field. As shown in many literatures, CGT has an easy axis in the out-of-plane direction. The remanence of the in-plane magnetization seems difficult to be justified.

A clue actually could be got from Fig. 3d and 3e, where Cooper pairs are found to be essential to the hysteresis. Is it possible that the Cooper-pair tunneling prefers the out-of-plane spin direction in the ferromagnetic layer whereas normal electrons do not? Such a picture could be also extended to Fig. 3a, in which the superconducting tunneling is suppressed by the in-plane magnetization at 100 mT. As such, it is the change of out-of-plane domains rather than the in-plane remanence that leads to the observed hysteresis. Combined with the mechanism depicted in Fig. 4d, a coherent discussion should be provided in the manuscript.

2. In Fig. 4a-b, a minimum is observed at fields shifted away from $B=0$ T. However, this does not need to be a signature of ϕ -JJ. In an asymmetric 0 - π JJ, this feature is commonly observed even if its ground state is 0 or π . An effective method is to follow the standard sweeping scheme shown in many works, e.g. PRL 109, 107002 (2012) and find the four critical current in the I_c -B phase diagram. Considering the importance of ϕ -JJ, such an experiment is strongly suggested.

By the way, the I_c -B pattern should change significantly as a function of the azimuthal angle of B fields, as long as there are strip domains in CGT. Especially when the field is perpendicular to the domain walls, it should restore to a Fraunhofer pattern. This is useful to exclude many possible scenarios and should be carried out.

3. The Fiske resonance is easier identified in the voltage (rather than current) driven experiment. As an important feature of the observation, is there more characterization in addition to Fig. S2?

4. For devices #2 and #3, the area of JJ should be presented, so as to support the conclusion that the multi-domain structures results from thickness rather than the large lateral dimension.

In addition, there are some unclear statements that need to be improved:

1. Line 164, 'the two nonequivalent S/F hetero-interfaces vertically'. Why is it related to the asymmetric values at $\pm B$ fields?

2. Line 169, 'still' is not necessary; 'at higher fields', not 'field'.

3. The derivation of the lower panel of Fig. 3a indicated by line 173 should be detailed in SI.

4. Line 242, the magnitude of coherence length of NbSe₂ should be presented.

5. Line 308, 'gained' is confusing.

6. Line 535, the magnetic structure should be added to the cartoon of CGT, and the pinned spin in NbSe₂.

Response to Reviewer's Comments

We acknowledge the reviewers for carefully reading our manuscript "Van der Waals Ferromagnetic Josephson Junctions" (NCOMMS-21-09874A) and for their insightful comments which help us improve the quality of our work. According to their kind advice, we have carefully reviewed and revised the presentation of our manuscript. We have also carried out new experiments to address the reviewers' concerns. The corresponding revisions in response to their comments have been made. The revised parts have been highlighted in the updated manuscript. The detailed revisions are listed on a separate page at the end of this response letter.

Response to Reviewer #1

General comment:

In this manuscript, Ai and co-authors fabricated Josephson junction built with vdW interfaces using NbSe₂ as a superconductor and Cr₂Ge₂Te₆ (CGT) as a ferromagnetic insulator. The author demonstrates the manipulation of Josephson critical current through the magnetism of CGT. The superconductor/ferromagnet/superconductor junction have received lots of interest, yet experiments were limited in the difficulty of fabrication of high quality devices using conventional deposition techniques. Here, Ai et al. show a possibility to fabricate high quality Josephson junction (that is confirmed by the observation of Fiske resonance) and observed cooperative phenomena between supercurrent and magnetism. The subject of this study can contain broad interest for solid state physics, therefore could be suitable for the aim of the submitted journal.

Although the results seem to be interesting, I found the discussion and interpretation of the results are not clear. There are quite a few points that discussions are rather hypothetical. I believe these points need to be significantly improved for the publication in the submitted journal. Please consider the following suggestions.

Response:

We thank the reviewer for his/her honest feeling on our manuscript and we appreciate the reviewer for the specific questions and suggestions on the presentation of experimental findings, which help us to further improve our manuscript.

In the following, we would like to address all the comments point-by-point and highlight the modifications made to the manuscript. We hope that our revised manuscript can remove the reviewer's concerns.

Comment 1:

In the analysis, author said $(d+2\lambda)$ is ~82 to 98 nm with d ~ few nm and London penetration depth, λ is few-tens of nm. The London penetration depth*

exceeds their NbSe₂ thickness (NbSe₂ is a few layers thick according to their method
section). It is not clear whether they can use this analysis when the penetration depth
exceeds the thickness of NbSe₂. Or did the author actually use a thickness of few-layer
NbSe₂ instead of lambda? Please add some explanation for this.

**Response:**

First, we're sorry to quantify the NbSe₂ flakes we used as few-layered samples,
because the thicknesses of which are around 20~40 nm and the superconducting
transition temperatures are close to bulk NbSe₂. Also, when the thicknesses of two
NbSe₂ electrodes (t_1 and t_2) are much smaller than their London penetration depth (λ_L
~ 0.1-0.3 μm [refs 1–3]), the effective junction length $d + 2\lambda$ should be corrected to⁴

$$d + \lambda_L \tanh(t_1/2\lambda_L) + \lambda_L \tanh(t_2/2\lambda_L),$$

and the result of it is about $d + 20\sim 40$ nm, where d is the thickness of Cr₂Ge₂Te₆
(CGT) barrier, from which we can get that the effective 2λ value is nearly identical to
the sum of the thicknesses of NbSe₂ electrodes. In device #01 mentioned in the main
text, the calculated 2λ is around 70~90 nm with the junction width $W \sim 5\text{-}6$ μm ,
however, the difference between the corrected result above may come from an
irregular shape of the junction area, leading to an inaccurate estimation of junction
width perpendicular to the magnetic field direction. Besides, we have defined the
periodicity of oscillation pattern ΔB from the first I_c minimum at 4.2 mT, but the
subsequent side-lobes have larger widths (the second I_c minimum at 9.8 mT and the
third one at 15.1 mT), which may presumably be due to a field focusing effect^{5,6}
caused by the nearby superconducting contacts. Therefore we may expect a wider
junction width W as well as a larger period ΔB , giving a more actual estimation of
calculated 2λ close to $\lambda_L \tanh(t_1/2\lambda_L) + \lambda_L \tanh(t_2/2\lambda_L)$.

**Comment 2:**

*I am concerned about the influence of the fringing field generated from the CGT.*
*Since CGT can become a multi domain structure, there is a fringing from the domain*
*wall. Please explain how authors can distinguish between the contribution of the*
*fringing field and the interpretation they provided in this manuscript.*

**Response:**

We thank the reviewer for pointing out the concerns on the influence of the
fringing field from the domain wall in the CGT, which we have not taken into
consideration before. By reexamining the data of hysteresis and carefully reading on
the related papers, we are so sorry to say that we could not give a precise quantitative
estimation of such an effect in our samples, though we still try to give a superficial
discussion on it.

It is known that the fringing field is the magnetic field emanating from a
ferromagnet, associated with the magnetic dipolar interactions equivalent to the
divergence of the magnetization within and at the surfaces of the sample⁷. The
formation of domain walls around magnetization reversal in the unsaturated state will
lead to a corresponding fringing field that does not cancel out by multiple domains of

different signs⁸. This field is far from being axially symmetrical because of the
 irregular shape of the crystal⁹, and also the amplitudes of it from the domain wall
 decay rapidly with distance on the order of hundreds of nanometers, both for in-plane
 and out-of-plane directions^{10,11}. Considering a very simple model^{11,12} in which the
 fringing field comes from a one-dimensional, linear, single domain wall within a thin
 ferromagnetic film, and in this geometry, the field strongly depends on the domain
 wall thickness parameter q as a function of the film's thickness $2b$ and its intrinsic
 magnetic properties including the exchange energy (J), the uniaxial anisotropy energy
 (K_u), and the saturation magnetization (M_s). For a given material like CGT, the q
 value can be numerically calculated by the self-consistent equation for a Bloch
 domain wall¹³ (which has been observed in CGT in recent reports¹⁴)

$$98 \quad \frac{J_x}{q^2} (\sqrt{2} - 1) = \frac{K_u}{2} + \pi M_s^2 \left[\frac{2q}{b} \log \left(1 + \frac{b}{q} \right) - \frac{q}{q+b} \right].$$

By taking the characteristic parameters of CGT obtained from several previous
 results ($J \sim 3.76$ meV [ref.15] and the consequent exchange stiffness $J_x \sim 1.53 \times 10^{-11}$
 J/m, $K_u \sim 3.3-3.7 \times 10^4$ J/m³ [refs 16–18] and $M_s \sim 2-3 \mu_B/\text{Cr}$ [refs 19–21]), the q value
 got at the film's thickness $2b \sim 6-7$ nm (same as the barrier thickness in device #01) is
 about 18-20 nm, similar to the domain wall width of the stripe type in CGT¹⁴. Then,
 the spatial distribution of the fringing field originated from such a domain wall can be
 derived by algebra expressions¹², and the influences calculated from the bottom of the
 ferromagnetic layer are presented in **Fig. R1(a)** and **(b)**. As shown in the figure, the
 maximum of the fringing field is up to several mT (taking $q = 18$ nm for a narrower
 domain wall by which the field strength is enhanced), in accordance with the
 hysteresis we got in our JJs, nevertheless, it decays rapidly away from the center of
 the domain wall. The general vertical size of our JJ device is smaller than 100 nm still
 within the range of a comparable fringing field (perpendicular to the film H_z), but the
 lateral size of it is about several μm that the fringing field (parallel to the film H_x) at
 the edge of interlayer CGT may have negligible impacts on the measured hysteresis in
 critical current or junction resistance, especially when the external magnetic field is
 applied along the in-plane direction. Also, the fringing field calculated in two thicker
 CGT films are plotted for comparisons (the q value is still in the range of 18-20 nm
 for these two thicknesses, so we also use 18 nm for simple estimations), from which
 we can read that the influence of the fringing field is more obvious in thicker samples.
 However, the hystereses obtained in our thicker barrier JJs (device #02 and #03) are
 smaller than that of the thinner one (device #01), which seems contrary to our
 expectation of a larger contribution of the fringing field to the hysteresis as the
 thickness increases.

**Fig. R1. Fringing field distributions calculated for CGT films with different**
 **thicknesses. (a)** The fringing fields from a Bloch wall for $q=18$ nm at the bottom of
 the film of a certain thickness (black for 6.5 nm, red for 11 nm, and blue for 20 nm)
 along the in-plane direction. **(b)** The fringing fields at the same conditions along the
 out-of-plane direction.

Besides, if the fringing field from the domains has significant contributions to
 the hysteresis, then we could expect a different result in the hysteretic Josephson
 coupling phenomenon whenever the temperature increases above the T_c of CGT and
 returns because the appearance of magnetic domains is in a random process which is
 unrepeatable. From the measurements of device #01 using two instruments, we did
 not find similar signatures both in the hysteresis and the oscillation patterns. To
 detailedly reveal the exact fringing field generated from the CGT needs spatially
 resolved intensity measurements that exceed our ability to realize at present, and so
 regrettable for us that we could not provide a confirmed quantitation of it. The model
 we used above is only a rough estimation to our knowledge on the fringing field from
 a single domain wall, as the calculated magnitude may be greatly different from the
 actual values, so it would be grateful if the reviewer could give us more suggestions
 on an improved model or some methods to evaluate this effect.

**Comment 3:**

*The following argument highly depends on the magnetic anisotropy constant and*
 *probably exchange stiffness and some other parameters of the ferromagnet. Authors*
 *need to justify this discussion by looking at literature or additional experiments.*

*“Considering our material systems, the equilibrium domain size of $Cr_2Ge_2Te_6$ is*
 *probably larger in ultrathin nanoflakes as mediated by a single-domain remanence, in*
 *contrast to the tendency of forming multi-domains when approaching a 3D bulk limit.”*
 *In addition to this, the magnetism of their CGT flake is not clear at all. Please add*
 *more detail characterization on the magnetism of the CGT material.*

**Response:**

We thank the reviewer for pointing out these important issues on the magnetic
 properties and domain structures of CGT. The birth of microscopic magnetic domain

structure in CGT is related to the competition between the magnetocrystalline
 anisotropy and magnetic dipolar interaction, where the latter one always prefers an
 in-plane magnetization and becomes significant in low-dimensional materials²². A
 multi-domain state is favored in thick ferromagnetic films due to the dipolar energy
 winning over the exchange energy²³, while in some cases for ultrathin flakes whose
 easy axis is perpendicular to the film, dipolar interaction is expected to be very small,
 probably leading to a very large domain size as shown in a six-layer CGT from **Gong**
 **et al's** results¹⁵. The dependence of the domain size (D) should both take the variety
 of domain wall energy (U_w) and also the film thickness (L) comparable with the range
 of exchange interactions (r_e) into account. The model put forward by **Kooy and Enz**²⁴
 predicted that the domain size is proportional to the film thickness when L is much
 larger than the characteristic 'dipolar length' (D_0), however as L is reduced below D_0
 while still larger than r_e , **Kaplan and Gehring**²⁵ found that the domain size will
 dramatically increase as film thickness decreases in the ultrathin regime. Taking the
 magnetic parameters of CGT as shown above, its domain wall energy density (σ_w) and
 the domain wall size can be estimated²⁶

$$174 \quad \sigma_w = 4 \sqrt{J_x(K_u - \frac{1}{2}\mu_0 M_s^2)}, \quad D = \pi \sqrt{J_x / (K_u - \frac{1}{2}\mu_0 M_s^2)}.$$

The domain size of the CGT is calculated to be in the range of about 60-90 nm, and
 the consequent dipolar length $D_0 = 2\sigma_w/\mu_0 M_s^2$ is around 3-13 μm , far greater than the
 thickness of ferromagnetic film L . Meanwhile, the range of exchange interactions in
 CGT is evaluated by the distance of neighboring Cr atoms, so the conditions required
 in the model above could be satisfied. Except for the theoretical estimations, we still
 want to discuss the single-domain or multi-domain structures in CGT nanoflakes or
 bulks observed in previous experiments.

In **Gong et al's** results¹⁵, the single-domain remanence is evident in ultrathin
 CGT ranging from 2-6L through Kerr rotation measurements, in contrast to the
 formation of multi-domains usually in the bulk. For comparison, the signatures of a
 multi-domain state have also been reported as the slanted Kerr rotation hysteresis
 loops of a 19 nm CGT sample from **Wang et al's** results²⁷, as well as the slanted
 induced Anomalous Hall Effect (AHE) signals in a 35 nm CGT/Pt device from
 **Lohmann et al's** results²³. For much thicker CGT samples (91-301 nm), **Han et al**¹⁴
 have observed stripe-like and bubble-like multiple domain structures assisted by the
 Lorentz transmission electron microscopy (LSTM). Also in the bulk CGT, **Guo et al**²⁸
 have found various multiple domain structures and symmetries stem from the
 competition between thermal fluctuation and anisotropy energy induced by magnetic
 dipolar interaction through magnetic force microscope (MFM) measurements. From
 the known observations of multi-domains in CGT nanoflakes or bulks, the
 characteristic domain size is around 70~120 nm, which is much larger than the
 few-layer CGT nanoflake thickness (<10 nm). To verify the magnetic properties of
 our CGT flakes with those reported, we have further performed some
 characterizations to give a comprehensive understanding.

[Redacted]

**Fig. R2. Domain structures in CGT flakes with different layer thicknesses. (a)**
Single domain remanence of an ultrathin 6L CGT acquired by Gong et al. (b) Slanted
Kerr rotation loops of a 19 nm CGT acquired by Wang et al. (c) Slanted AHE signals
of a 35 nm CGT/Pt device acquired by Lohmann et al. (d) Stripe-like and bubble-like
multiple domain structures of thick CGT (91-301 nm) acquired by Han et al. (e)
Various symmetries of multiple domains of bulk CGT acquired by Guo et al.

We first characterized the intrinsic magnetic properties of CGT bulk crystals,
where the magnetization measurements are performed through a SQUID
magnetometer. The Curie temperature $T_c \sim 63$ K was determined from the drop in the
temperature dependence of its magnetization when a 100 mT magnetic field is applied
to the in-plane (along a-b plane) and the out-of-plane (along c axis) of the crystal, as
shown in **Fig. R3(a)**. The bulk magnetizations below T_c with an external magnetic
field reveal a ferromagnetic behavior with magnetic anisotropy by the comparison of
their saturation fields measured at 2 K, as shown in **Fig. R3(b)**. With a larger
saturation field along the in-plane direction, the CGT crystal shows an easy axis
perpendicular to the a-b plane, in agreement with previous reports^{16,29}.

**Fig. R3. Magnetic characterizations of bulk CGT by SQUID magnetometer. (a)**
 Temperature dependence of the bulk CGT sample's magnetization with a 100 mT
 magnetic field applied along the out-of-plane (black) and in-plane (red) directions. **(b)**
 Field dependence of its magnetizations at 2 K (black curves for out-of-plane direction
 and red curves for in-plane direction).

To further confirm the magnetic behavior in exfoliated CGT nanoflakes, we
 performed the Kerr rotation measurements to explore their properties changing with
 layer thickness. The measured temperature is at 5 K, below the critical temperature of
 NbSe₂ which is used as a superconducting electrode in our van der Waals (vdW)
 Josephson junction (JJ) devices. The original Kerr rotation signals under the
 out-of-plane magnetic field are present in **Fig. R4(a)**, where we can obtain that the
 saturation magnetizations of CGT nanoflakes increase as the film thicknesses increase.
 In **Fig. R4(b)** we present the normalized curves extracted from **Fig. R4(a)** for clarity
 in comparison, and we can clearly see that for thinner CGT samples like 4 nm and 12
 235 nm flakes, they only show monotonic transitions in the Kerr rotation hysteresis loops.
 Besides, the magnetic anisotropy is more obvious in the thinnest 4 nm CGT. Yet a
 slanted behavior, as well as multiple transitions from the Kerr rotation loops under
 lower magnetic field, appears in the 20 nm CGT sample, suggesting the formation of
 nonuniform magnetizations in this thickest CGT flake, probably arisen from such as
 multi-domains.

**Fig. R4. Magnetic characterizations of nanoflake CGT by MOKE. (a)** Kerr angle
measured at 5 K of three CGT nanoflakes with different thicknesses (black: 4 nm, red:
12 nm, blue: 20 nm). **(b)** Renormalized Kerr angle obtained from the data in **(a)**
(stacked for clarity).

Combined the results reported before with our characterizations on CGT samples,
both the larger domain size up to 10 μm as a single-domain remanence in ultrathin
nanoflakes and the smaller domain size around 100 nm along with a multi-domain
structure in much thicker CGT samples or even bulk crystals have been confirmed.
Although we haven't found the counterexamples in these two thickness regions for
CGT, we cannot conclude a solid correlation between the equilibrium domain size of
CGT with its thickness, nor could we prove the tendency of forming multi-domains as
the film thickness increases approaching a 3D ferromagnet up to our considerations.
The basic purpose to give such an argument is trying to explain the enhancement of
Josephson current in the thick-barrier device, therefore we will only limit our
discussions to the range of CGT thickness below 20 nm according to our experiments.

**Comment 4:**

*The magnetic field-induced hysteresis author discussed in Fig.2 and Fig.3 is a*
*rather small value and definition of exact zero magnetic field can be difficult. How*
*did the author make sure that this is not the artifact of measurement instruments such*
*as the remnant field of the coil?*

**Response:**

We thank the reviewer for the concerns on the magnitude of hysteresis of our JJs.
The magnetic hysteresis determined from our Josephson current measurements is
around several mT using few-layer CGT as tunneling barriers, which is proportional
to the remanence of its in-plane magnetization. We do believe that this value is
trustworthy and independent of the definition of exact zero magnetic fields in any
measurement system. First, the magnetic hysteresis given by our device is at the same
magnitude as that of NbN/GdN/NbN JJs [refs 30–32] (1~5 mT), proving that such a
hysteresis value is not too small to be detected. Second, all the three devices in the
Josephson coupling regime presented in the main text are of different values of
hysteresis (device #01: ΔB - 5.4 mT when B_{max} is 30 mT, ΔB - 6.5 mT when B_{max} is
50 mT; device #02: ΔB - 3 mT when B_{max} is 30 mT; device #03: ΔB - 2 mT when B_{max}
is 50 mT), suggesting the intrinsic dependence on the barrier's magnetism. To further
confirm that it is not from the artifact of the measurements instruments, we have
carried out a series of comparative experiments and the results are as follows.

At first, to directly exclude the influences from the remnant field of the coil, we
have fabricated nonmagnetic JJ with NbSe₂ in the vertically stacking configuration.
Before the fabrication, the unilateral NbSe₂ on the substrate was slightly oxidized in
the ambient environment and then transferred into the glove box for subsequent
operations, which leads to the relatively large junction resistance (R) and small critical
current (I_c) in device #05, measured in a normal PPMS instrument (PPMS-1) down to

1.9 K. The contour plots of junction differential resistance (dV/dI) as a function of
 bias current (I) and in-plane magnetic field ($B//$) upon two sweeping directions are
 shown in **Fig. R5(a)** and **(b)**. Distortions of the I_c patterns are due to the self-field
 effect^{33,34}, inducing that the positions of maximum I_c peaks are not at zero magnetic
 field. The comparison of two sweeping branches at the base temperature shown in **Fig.**
 **R5(c)** gives a nearly zero hysteresis value at the field applied step of 0.5 mT. Within
 the detection accuracy, the remnant field of the magnetic coil has no impact on the
 Josephson current measurements.

**Fig. R5. I_c patterns in non-magnetic Josephson junction device #05.** (a-b) Contour
 plots of junction dV/dI as a function of I and in-plane B upon two sweeping directions,
 measured in a normal PPMS at 1.9 K. Left is from positive to negative field and right
 is from negative to positive field. (c) Extracted I_c values compared in two branches,
 indicating a zero hysteresis.

Then, to prove that the field-induced hysteresis of vdW magnetic JJ is
 304 determined by the barrier itself rather than the errors in measurements, we also used
 another vdW ferromagnetic insulator CrBr₃ to serve as the tunnel layer. The JJ device
 #06 was fabricated through a similar dry-transfer method with a 4 ± 0.8 nm CrBr₃
 nanoflake, and its characterizations were carried out in another PPMS instrument
 (PPMS-2) equipped with a dilution refrigerator (DR) down to 0.1 K. As above, the I_c
 patterns at two sweep directions of the device are present in **Fig. R6(a)** and **(b)**. The
 hysteresis of I_c at base temperature is given around 1 mT with the field applied step of
 0.5 mT and B_{max} of 42 mT (**Fig. R6(c)**), which is smaller than that of the CGT-JJ
 device. The small remanet magnetization may be owing to the weak uniaxial
 anisotropy of CrBr₃ with soft ferromagnetism³⁵⁻³⁷, but also its thickness should be
 taken into account.

**Fig. R6. I_c patterns in magnetic Josephson junction device #06 based on CrBr₃.**
 **(a-b)** Contour plots of junction dV/dI as a function of I and in-plane B upon two
 sweeping directions, measured in another PPMS equipped with a DR at 0.1 K. Left is
 from positive to negative field and right is from negative to positive field. **(c)**
 Extracted I_c values compared in two branches, indicating a hysteresis of 1 mT due to
 the ferromagnetic CrBr₃.

Finally, return to device #01, we both measured it in these two PPMS
 instruments to obtain the full-range temperature dependence of its hysteresis, and the
 comparison between them is shown in **Fig. S3(c)** (the same as **Fig. R7(a)** here) from
 the supplementary materials. From the arrows indicated we can see the central
 positions of two sweep branches of its in-plane magnetoresistance are slightly shifted,
 to the same direction with an almost identical deviation. The field-sweeping speed is
 0.1 mT per second, which provides sufficient data points to identify the hysteresis.
 The hystereses got at two instruments for the same device are very close to each other.
 Although the temperatures are varied for these two situations, we have previously
 shown that the hysteresis of device #01 becomes likely saturated below 2 K in the
 main text **Fig. 3(d)** (the same as **Fig. R7(b)** here). Also, the exact zero magnetic field
 may not be located at the zero position as we read, yet it does not affect the observed
 intrinsic magnetic hysteresis of the barrier. At last, we summarize the hystereses of
 different devices from different measurement instruments in **Table. R1** for clarity.

**Fig. R7. Transport properties of device #01 in two measurement instruments. (a)**

Comparison of hysteretic resistances obtained in these two measurement systems in
 different temperature ranges (1.9 K in the normal PPMS and 0.1 K in the PPMS
 equipped with a DR). (b) Temperature dependence of the hysteresis from 0.1 K to 7 K
 in the combination of the data got in two instruments.

Table 1

Number	Interlayer	Instrument	ΔB (mT)	B_{max} (mT)
device #01	$\text{Cr}_2\text{Ge}_2\text{Te}_6$	PPMS-1 (1.9 K)	5.6	27
		PPMS-2 (0.1 K)	5.4	30
device #02	$\text{Cr}_2\text{Ge}_2\text{Te}_6$	PPMS-2 (0.1 K)	3	30
device #03	$\text{Cr}_2\text{Ge}_2\text{Te}_6$	PPMS-2 (0.5 K)	2	50
device #05	— (NbSe_2)	PPMS-1 (1.9 K)	0	30
device #06	CrBr_3	PPMS-2 (0.1 K)	1	42

Table. R1. Summary of the hystereses for the devices mentioned above.

Comment 5:

 *In Fig. 3(d) and 3(e), the author demonstrates the temperature and current*
 *dependence of the results. Is it possible that the superconductivity (or Josephson*
 *junction) is sensitive to probe the phenomena that makes observables below T_c of*
 *NbSe_2 , yet phenomena itself still remain above T_c but there isn't enough sensitivity to*
 *detect it above the T_c .*

Response:

 We agree with the assumptions on the sensitivity to probe the hysteretic
 phenomena through the Josephson current below the T_c of NbSe_2 electrodes, which
 can become less observable when it is non-superconducting. As we have mentioned in
 the main text (starts from line 181), we first observed that the hysteresis shrinks
 apparently both in junction resistance (R) amplitude and magnetic field range as
 temperature increases, and finally disappears above T_c at 6.5 K.

For a 2D vdW ferromagnet, the Curie temperature is strongly dependent on its
 layer thickness in the ultrathin regime because of the dimensionality effect¹⁵. From
 **Gong et al's** results¹⁵, a six-layer CGT (4.6 nm) shows an intrinsic transition

temperature close to 10 K, while the thickness of the CGT barrier in our JJ device #01
 is about 6.5 nm, supporting that the CGT we used here is still in the ferromagnetic
 state. Next, we found a dramatic decrease of hysteresis above 5.5 K, accompanied by
 the fact that the discrepancies between the two sweep branches from 6-6.5 K become
 illegible to distinguish. The R varies little within the range of the applied magnetic
 field. Besides, similar phenomena were also found at the high dc-bias conditions
 when superconducting electrodes were driven into the normal state.

Based on the experimental observations above, we intend to assume that the
 hysteresis can denote the magnitude of Josephson interference in such a specific JJ
 structure. With temperature increasing, thermal excitation becomes significant and the
 phase coherence is suppressed. The possibility of the reduction of Josephson
 penetration depth with temperature increasing could contribute to it. Comprehensively,
 we would like to compare this with the tunnel junction structure's result with a thicker
 CGT barrier, in which we found that the hysteresis in resistance of device #04 (14.5
 380 nm) also vanished above T_c , as shown in **Fig. R8(a)**. In the latter structure, the
 381 transport conductance is dominant by the quantum tunneling effect through
 single-electron channels, completely different from the mechanism of a JJ. It should
 be mentioned that neither of these two devices (#01 and #04) shows observable
 magnetoresistance effect at 6.5 K, at which temperature they could both have been
 treated as conventional magnetic tunnel junctions consisting of normal metals with
 ferromagnetic insulators. This may be attributed to the very weak anisotropy of CGT
 in its ab -plane¹⁸. Meanwhile, the variation range of resistance in device #04 is larger
 at 2 K than that of at 4 K, as the zero-field junction resistance grows increasingly
 when cooling down to the lowest temperature (**Fig. R8(b)**), which is another support
 to the necessity of superconductivity in probing the hysteretic phenomena.

 **Fig. R8. Transport properties of a tunneling junction device #04. (a)** In-plane
 magnetoresistance of device #04 at 2 K, 4 K, and 8 K compared among the
 superconducting state and normal state. **(b)** Temperature dependence of this tunnel
 junction resistance at zero magnetic field.

 In summary, we conclude that superconductivity is an effective way to probe the
 hysteresis in our devices because of its sensitivity, owing much to the obvious

changes of the junction resistance below the T_c of NbSe₂. We still want to argue that
the validity comes from the superconducting correlations between two NbSe₂ through
a CGT barrier rather than merely accounting for the transitions of NbSe₂ themselves,
both for the Josephson coupling and superconducting tunneling regime.

**Comment 6:**

*The section named “Evidences of π (π) phase coupling” does not contain a clear*
*explanation to justify their interpretation and need to be improved. I am not sure that*
*the presence of π (π) phase coupling is justified in the experiment. For example, the*
*asymmetry in the current can be possible with Oersted field contribution on the*
*Josephson junction measurement (please see page 57 of*
*http://www.wmi.badw.de/teaching/Lecturenotes/AS/AS2013_Chapter2_Slides.pdf).*
*Maybe the author requires phase sensitive measurement on the Josephson critical*
*current or makes correlation between the real magnetic structure of their CGT in the*
*junction and transport data by some means.*

**Response:**

We thank the reviewer for raising this essential question for our experiments and
allowing us to add more evidence to support the π phase coupling. In this section
named ‘Evidence of π phase coupling’, we intend to attribute the observations of
non-Fraunhofer pattern with a central minimum of critical current (I_c) to the
consequence of a likely 0- π JJ structure, as we have excluded the influence of
Abrikosov vortex which may also lead to a similar signature in the I_c patterns. Origin
of the π ground state phase through the CGT barrier is worthy of discussion, and we
have put forward two possible scenarios to explain: one is from the
momentum-conserving tunneling process of the spins in CGT which generates a
π -shift to the supercurrent^{38,39}, the other is from spin-mixing effects at the magnetic
inhomogeneous S/F interfaces with nonuniform exchange interactions, where the
spin-dependent scattering phases are close to π ⁴⁰⁻⁴⁴. We have to admit that we did not
reveal the actual physical process conclusively in our JJ experiments, and the
existence of the π phase is not so persuasively merely supported by the
non-Fraunhofer patterns. Therefore we have carried out the phase-sensitive
measurements using a SQUID structure^{45,46} to define the ground state phases in
NbSe₂/CGT JJ systems, as advised by the reviewer.

It is known that for a conventional JJ the ground state phase equals 0, while in an
SFS JJ with a properly thick F-layer a π phase can be realized. To go a step further, a
JJ with an arbitrary value φ of the phase between 0 and π named φ -JJ possesses
doubly degenerate ground states, which is of great potential to be utilized as a
component for superconducting quantum devices⁴⁷⁻⁵⁰. To detect the phase shift in our
systems, we have constructed the SQUIDs consisting of an SFS JJ (NbSe₂/CGT) and
a reference JJ (NbSe₂/NbSe₂). Details of the fabrication process can be got in
**Methods. Fig. R9(a)** displays the false-color SEM image of SQUID #07, and its
transport results as SQUID oscillations in critical current at 1.85 K are presented in
**Fig. R9(b)**. The SQUID critical current as a function of magnetic flux Φ is counted

through the SQUID loop ($17.5 \mu\text{m}^2$) as the sum of the empty areas ($\sim 5.9 \mu\text{m}^2$) and the
 screening areas ($\sim 11.6 \mu\text{m}^2$), agreed with the area calculated from the oscillation
 period ($\sim 0.118 \text{ mT}$), shown in **Fig. R9(c)**. To extract the phase of the SQUID #07,
 we have calibrated the measurement systems through a NbSe₂ SQUID on the same
 chip to give zero magnetic field, the data of which is also plotted in **Fig. R9(c)** for
 comparison. Under this condition we can find that the position of maximum critical
 current in one periodicity of SQUID #07 is deviated from zero magnetic field (guided
 by the red line), providing a nontrivial phase $\varphi = 148.6^\circ$ rather than a 0 or π phase.

 **Fig. R9. Josephson phase measurements of SQUID #07.** (a) False-color SEM
 image of SQUID #07 (scale bar: $2 \mu\text{m}$), consisting of an SFS JJ with a CGT barrier
 and an intrinsic reference JJ formed by two NbSe₂. The area of the SQUID loop (17.5
 457 μm^2) is the sum of the empty parts ($\sim 5.9 \mu\text{m}^2$) and the screening areas ($\sim 11.6 \mu\text{m}^2$). (b)
 Differential resistance dV/dI (log-scale) map of SQUID #07 as a function of bias
 current I and out-of-plane magnetic field measured at 1.85 K , showing a SQUID
 oscillation. All the dV/dI curves are sweeping from zero to positive bias. The position
 of zero magnetic field calibrated by a NbSe₂ SQUID on the same substrate chip is
 indicated by the purple line in this plot. (c) Critical current I_c oscillations of SQUID
 #07 (black rectangles) with the period of 0.118 mT , consistent with the SQUID loop's
 area. The phase φ of it is about 148.6° , obtained by a fit to I_c with the sine relation
 (black lines). The red circles show the I_c oscillations of the NbSe₂ SQUID for
 calibration with the period of 0.209 mT . (d) Temperature dependence of I_c of SQUID
 #07 at zero magnetic field. The thickness of the CGT barrier is $\sim 3.5 \text{ nm}$ measured by
 AFM. The nonmonotonic change of I_c implies an incomplete $0-\pi$ transition in SQUID
 #07 with a φ ground state phase at low temperature.

The single nontrivial phase obtained in our NbSe₂/CGT SQUID is probably

originated from the ground state phase of a ϕ -JJ, nevertheless its Josephson energy
profile as a double-well potential should give doubly degenerate ground states phases
$\pm\phi$. Upon applying a bias current to the JJ corresponding to a tilt of the potential, the
phase could be trapped either in the $+\phi$ or $-\phi$ state following a specific bias sweeping
sequence^{51,52}. When switching the JJ from the zero-voltage state to the voltage-state,
the required tilt for the escape of the phase out of the potential well depends on the
initial state ($+\phi$ or $-\phi$) of the junction, at a higher I_{c+} or a lower I_{c-} alternately as we
could get in positive bias currents. **Goldobin et al**^{48,50,52} have demonstrated the
possibility to use $I_{c\pm}$ for the readout of the $\pm\phi$ state. However, the I_{c-} can only be
observed together with the I_{c+} for low damping (underdamped JJ) allowing a
stationary phase motion^{48,53}, because the probability to find the phase trapped in a
certain state is nearly constant for $\pm\phi$. In contrast, for the regime of high damping
(overdamped JJ), the retrapping process is deterministic to predict the particular
destination well of the Josephson phase, therefore only the higher I_{c+} can be measured.
Due to non-negligible electronic noises in the external circuits of our instruments as
we have suggested in the main text, we were not able to evaluate the damping in our
JJs accurately^{54,55}, since their IVCs show tiny hystereses in the process of switching
and retrapping like high damping JJs. As a result, we only observed one of the ground
state phases $\phi = 148.6^\circ$ corresponding to the higher I_{c+} in our experiments. Although
the lower I_{c-} for the other phase is not capable to realize in our high damping JJs, the
nontrivial phase (not 0 or π) of SQUID #07 could still be regarded as evidence to
support our assumptions on the π phase coupling.

To further address the ground state phase, we have measured the critical current of
SQUID #07 in the variation of temperature, and surprisingly found a
non-monotonic dependence which is likely to be an induced $0-\pi$ transition occurring
at $T \sim 0.8 T_c$ in this device, as shown in **Fig. R9(d)**. The thickness of the CGT barrier
is about 3.5 ± 0.7 nm measured by AFM. Compared with metallic ferromagnetic π -JJ
which requires the thickness of the barrier to be comparable to the characteristic
length of the order-parameter's damped oscillation, the decay length of insulator CGT
was measured by **Idzuchi et al**³⁹ to be around 1.4 nm, much thinner than the barriers
we used within 3.5~11 nm. The unexpected long-range correlations existing in such
thick ferromagnetic JJs might imply the presence of spin-triplet supercurrents⁵⁶⁻⁵⁸,
where the magnitude of the critical current decays much slowly with barrier thickness
increasing. This phenomenon has been extensively studied in metallic ferromagnetic
JJs, while **Caruso et al**⁴⁴ first discovered the signatures of spin-polarized triplet pairs
in the insulating-barrier JJs NbN/GdN/NbN. According to their results, nonuniform
exchange interactions, as well as spin-triplet correlations, play an important role to
explain the incomplete $0-\pi$ transition they observed in the temperature dependence of
the critical current $I_c(T)$, different from the complete transition due to the presence of
homogeneous exchange fields inside a metallic ferromagnet. Based on the assumption
of interfacial inhomogeneities with high spin-filter efficiency of the barrier and a large
spin mixing angle between the different transport channels, stronger magnetic activity
is expected to be dominant in longer JJs, in which the deviations from the
conventional Ambegaokar Baratoff (AB) behavior are more evident for thicker barrier

samples as the experimental data. Moreover, their recent investigations^{59,60} showed
how the spin-orbit coupling (SOC) interaction as spin-mixing mechanisms and on-site
magnetic impurities affect this unconventional $0-\pi$ transition, essentially due to the
variation of relative amplitudes of different spin correlation functions. Tuning the
strength of SOC or the impurity potential can stabilize the system into the 0 or the π
energy state over the whole range of temperatures.

When referring to our results, the shape of the $I_c(T)$ curve in SQUID #07 is a
more standard cusp-like $0-\pi$ transition with a higher weight of the s-wave singlet
component, corresponding to the relatively low values of disorder and spin-mixing
effects. Since the origin of a ϕ ground state phase comes from the lateral combination
of 0 -JJ and π -JJ parts coexisting in the same junction region, either of them may
undergo the $0-\pi$ or $\pi-0$ transitions at different temperatures, therein we are not able to
determine the phase at higher temperature to be either at 0 or π . Anyway, we believe
the observations above are convincing enough to support our interpretation of the
authenticity of π phase coupling in our NbSe₂/CGT JJs. Combined with the evidence
addressed previously as the non-Fraunhofer patterns, to explore how the ground state
phase as well as the spin-orbit interaction or impurity potential evolves with barrier
thickness, and also to take into account the realistic micromagnetic nature of barriers
like multi-domain structures, worths further investigations on the regime adapted for
full-range thickness in such a ferromagnetic insulator JJ system.

**Comment 7:**

*I was assuming that the author uses a few-layer thick NbSe₂ throughout their*
*manuscript according to their method section. However there is a sentence “bi-lateral*
*few-layer NbSe₂ (20 ~ 40 nm)” in supplementary information, which seems to be*
*confusing. Please confirm all the thickness of NbSe₂ and specify in each device.*

**Response:**

We thank the reviewer for point out our incorrect definition of the NbSe₂ flakes
we used as few-layered samples. A few-layered vdW material is generally defined as
a 2-6L sample which is thinner than about 5 nm, while the thicknesses of the NbSe₂ in
our JJs are around 10~40 nm obtained by AFM. The superconducting properties of the
electrodes above are close to the bulk NbSe₂ from the values of T_c and critical
current density. The specific thicknesses of NbSe₂ in each device (consisting of JJs,
tunnel junctions, and SQUIDs) are listed below in Table. R2.

Table 2

Number	NbSe ₂ (top) / nm	NbSe ₂ (bottom) / nm	Interlayer / nm	Barrier Material
JJ #01	20 ± 2	9.5 ± 0.5	6.5 ± 0.4	Cr ₂ Ge ₂ Te ₆
JJ #02	35 ± 1	23 ± 2	10.5 ± 0.3	Cr ₂ Ge ₂ Te ₆
JJ #03	34 ± 3	32 ± 1	11 ± 0.5	Cr ₂ Ge ₂ Te ₆
Tunnel #04	17 ± 0.5	21 ± 1	14.5 ± 0.4	Cr ₂ Ge ₂ Te ₆
JJ #05	25 ± 0.5	17 ± 1	—	— (NbSe ₂)
JJ #06	32 ± 2	14 ± 0.5	4 ± 0.8	CrBr ₃
SQUID #07	38 ± 1	13 ± 1	3.5 ± 0.7	Cr ₂ Ge ₂ Te ₆

Table. R2. Summary of the thicknesses of NbSe₂ in all devices.

Comment 8:

Since they have data for CGT thickness dependence of junction's temperature dependence, I think they can also check the CGT thickness dependence of junction RA(resistance area product). Please include this information which may further support the change of transport property change with the thickness of CGT.

Response:

We thank the reviewer for the suggestions on verifying the CGT thickness dependence of useful parameters. It is known that the transport property of the tunnel junctions can be obtained from the thickness dependence of junction resistance area product $R_n A$, and the result of our experiments is shown in **Fig. R10(a)** containing all the CGT devices from the Josephson coupling state to the tunneling region as defined in the supplementary information. Samples within the thinner barrier region but still with residual resistance are also included. The normal state resistances R_n of all the junctions were measured at 8 K above T_c . We find an exponential increase of $R_n A$ fitted to an $\exp(d_F/t)$ with the barrier thickness d_F ranging from 6-15 nm (guided by the red dashed line), where the characteristic quasiparticle tunnel length t is calculated to be 2 nm, slightly larger than the value obtained by **Idzuchi et al**³⁹ of 1.3 nm. However, the normalized barrier resistance constant is around 3-4 $\Omega \cdot \mu\text{m}^2$, which is much lower than their result of 340 $\Omega \cdot \mu\text{m}^2$. The relatively small value of our junctions may partially be attributed to the decrease of the semiconducting gap of thicker CGT than the ultrathin flakes as 1-6L they used, yet, the leakage of charge carriers across the barrier is more likely to be the main cause. Furthermore, we summarize the thickness dependence of the $R_n A$ product, the critical current density J_c and the $I_c R_n$

product of the only JJs whose resistances have exactly dropped to zero at 2 K in our
 PPMS instruments. As in **Fig. R10(b)**, except for the thickest barrier sample with an
 enhanced supercurrent mentioned before in the main text, we find that the J_c
 decreases with increasing d_F in the rest devices, but the $I_c R_n$ product does not show a
 monotonic dependence. Besides, neither the $R_n A$ product nor the J_c shows an
 evidently exponential change as expected for an effective insulating barrier, leading to
 the moderate variations of the $I_c R_n$ product. The relatively slow decrease of the J_c can
 also support the assumption of charge leakage to explain why our CGT barrier is not
 so insulating as it should be. Finally, we have to admit the individual differences in
 our samples reflected in their transport properties, as it is hard to control all the
 variables in the fabrication process of these vdW JJs, and we are also inclined to
 ascribe some observations above to an unphysical excuse. At the same time, the data
 based on these 5 JJs are not adequate to entirely reveal the superconducting nature
 beneath, therefore more detailed experiments are needed especially for the ultrathin
 CGT barriers in further studies.

 **Fig. R10. Barrier thickness dependence of the parameters.** (a) The resistance area
 $R_n A$ product of the junction (black rectangles) with different barrier thicknesses. The
 fitting result of the data by the exponential relation is indicated by the red dashed line.
 (b) The $R_n A$ product (black rectangles), critical current density J_c (red circles), and
 characteristic voltage $I_c R_n$ product (blue triangles) of the junction with different
 barrier thickness for absolute JJs whose resistances drop to exactly zero at 2 K.

**Based on the reviewers' comments, we have performed the following corrections:**

 **Revised manuscript:**

- 1. Page 1: We have revised the **Abstract** by adding a sentence to include the SQUID
 results we have finished afterward.
- 2. Page 3: We have revised the **Introduction** by adding up the SQUID results.
- 3. Page 6-7: We have added the descriptions of our new results of SQUID #04 to

jointly support the ‘**Evidence of π phase coupling**’ in our CGT JJs with the
unconventional Fraunhofer patterns.

4. Page 7-8: We have rearranged the interpretations of trapped vortex and
asymmetries in ferromagnetic interlayers from ‘**Evidence of π phase coupling**’ to
**Discussion** for a better organization. Also, we have added one sentence to explain
why we do not merely consider the contribution of the possible Oersted field
effect in our results of the asymmetry in critical current.

5. Page 8: We have revised the controversial statement on the correlation between
the equilibrium domain size and the film’s thickness in CGT by softening the
certain tendency towards the formation of multi-domains, and more necessary
explanations are given in the **Supplementary Material**.

**Locally generated** Oersted field

**Revised methods:**

1. Device fabrication: We have deleted the description ‘Few-layered’ to avoid
contradiction with the thicknesses of NbSe₂ electrodes (10-40 nm).

2. Device fabrication: We have added some details to the fabrication process
including the SQUIDs.

**Revised figure and table:**

1. Fig. 4: We have changed the title of its figure caption for a more explicit
description.

2. Fig. 5: We have added the results of Josephson phase measurements to further
support the assumptions on the π phase coupling in our CGT JJs.

**Revised supplementary material:**

1. Supplementary Section V: We have added the discussions of the domain size in
CGT and some basic magnetic characterizations on it for a more comprehensive
understanding in this part.

2. Supplementary Section VI: We have added the barrier thickness dependence of
the parameters including the $R_n A$ product, the critical current density J_c , and the
characteristic voltage $I_c R_n$ product, which might be helpful to reveal the transport
properties in our devices.

**Response to Reviewer #2**

**General comment:**

*In the manuscript of L. Ai et al, a van der Waals type of magnetic Josephson*
*junction is constructed by stacking superconducting NbSe₂ and ferromagnetic*
*insulator CGT. The transparent interface allows for the observation of a complete*
*Fraunhofer pattern and the associated Fiske resonance. Detailed characterization of*
*hysteresis are conducted as a function of in-plane magnetic fields, bias current and*
*temperatures. In particular, for a thicker ferromagnetic layer, the JJ exhibits a dip of*
*critical current around zero Tesla, which is claimed to be a signature of phi-JJ.*

*The devices seems to be of high quality and the measurement results are in*
*details. Nevertheless, there are several issues to be clarified.*

**Response:**

We thank the reviewer for the careful reading of our manuscript and we
appreciate the reviewer for the positive and encouraging comments on our work.

In the following, we would like to provide detailed answers to all the issues
addressed, and the corresponding revisions have been implemented in the revised
manuscript. We hope that our revised manuscript can remove the reviewer's concerns.

**Comment 1:**

*In Fig. 2-4, a striking hysteresis is observed upon sweeping the in-plane B field.*
*As shown in many literatures, CGT has an easy axis in the out-of-plane direction. The*
*remanence of the in-plane magnetization seems difficult to be justified.*

*A clue actually could be got from Fig. 3d and 3e, where Cooper pairs are found*
*to be essential to the hysteresis. Is it possible that the Cooper-pair tunneling prefers*
*the out-of-plane spin direction in the ferromagnetic layer whereas normal electrons*
*do not? Such a picture could be also extended to Fig. 3a, in which the*
*superconducting tunneling is suppressed by the in-plane magnetization at 100 mT. As*
*such, it is the change of out-of-plane domains rather than the in-plane remanence that*
*leads to the observed hysteresis. Combined with the mechanism depicted in Fig. 4d, a*
*coherent discussion should be provided in the manuscript.*

**Response:**

We thank the reviewer for the arguments on the origin of hysteresis we got in our
experiments. After a careful re-analysis of our results, we prefer to insist that the
observed hysteresis here is the consequence of the barrier's in-plane remanence due to
the following interpretations.

In the simple case of a homogeneous ferromagnetic barrier, when the barrier
magnetism is parallel to the applied magnetic field B , the Fraunhofer pattern is simply
offset and is hysteretic when B exceeds the coercive field B_c , yet the width of the
hysteresis in $I_c(B)$ is not B_c but is instead proportional to the saturation magnetization
M_s of the ferromagnet⁶¹. According to the Cr₂Ge₂Te₆ (CGT) barrier which has an easy

axis perpendicular to the cleaved plane (both from the previous studies and our
magnetic characterizations), we find that the coercivity along the out-of-plane
direction at zero magnetizations is relatively larger than that along the in-plane
direction in the case of the bulk CGT (**Fig. R11(a)**). When the thickness is down to a
few layers, we have used Kerr measurements to study its magnetic properties as B_c
along the out-of-plane direction to give a comparison with the hysteresis observed
upon sweeping the in-plane B in our magnetic Josephson junctions (JJs). We chose a
4 nm thick CGT flake whose thickness is close to that of in device #01 (~ 6.5 nm) and
another 12 nm sample that is near to the barrier thickness in device #02 (~ 10.5 nm)
and device #03 (~ 11 nm). The results are presented in **Fig. R11(b)**. For the 4 nm thick
CGT, the saturation field B_s in the perpendicular direction is around 30 mT, smaller
than the in-plane B_s of 75 mT as the result of device #01, while the extracted $2B_c$ of
13 mT is larger than the hysteresis ΔB of device #01 of 6.7 mT, in both of which the
maximum applied B starts from 50 mT. For the 12 nm thick CGT, its $2B_c$ value is 11
mT with the B_s of about 50 mT, which is also larger than the ΔB of around 3 mT and
2 mT, respectively obtained from the $I_c(B)$ curves of device #02 and #03. The relative
differences imply that the ΔB in $I_c(B)$ is more likely to reflect the remanence of the
in-plane magnetization of CGT, rather than its intrinsic coercivity.

**Fig. R11. Characterizations of the coercivities and saturation fields of bulk and**
**nanoflake CGT. (a)** Field dependence of the bulk CGT magnetizations at 2 K
measured by SQUID magnetometer (black for out-of-plane direction and red for
in-plane direction). Right inset is its zoom-in plot near the zero magnetic fields,
from which the coercivity force $2B_c$ along its easy axis is obtained as 34 mT, much greater
than the other value along its hard axis. **(b)** Kerr angle measured by MOKE at 5 K
of two CGT nanoflakes with different thicknesses (black for 4 nm and red for 12 nm).
The out-of-plane coercivities of them are 13 and 11 mT, respectively.

Also, we may not be able to conclude that the evident hysteresis observed is
directly related to the preference between the Cooper pair tunneling process and the
specific spin direction of the barrier. First, the hysteretic phenomena above the T_c of
NbSe₂ are less observable due to the lack of sensitivity to probe them. The reduced
variations of the junction resistance in the normal state restrict our ability to confirm

the ferromagnetism of CGT at that temperature. Besides, if the Cooper pairs are more
preferable to the spinful barrier whereas the normal electrons are not, then one may
expect that the transport properties will be presumably dependent on the proportions
of the carriers above from the NbSe₂ electrodes. However, from **Fig. 3(e)** in the main
text, we can find that the magnitude of the hysteresis remains almost the same, rather
than a progressive decrease as the d.c. bias current increases till its critical current.
Meanwhile, we tend to believe that the suppression of I_c at 100 mT as shown in **Fig.**
**3(a)** is attributed to the relatively large magnetic field rather than the in-plane
magnetization. This can also be supported by the comparison of maximum I_c in the
central as well as the second lobe of the Fraunhofer patterns under different
magnetized conditions. From **Fig. R12(a)**, the position of the maximum I_c shifts as
the sweeping process starts from a higher negative magnetic field, while the value of
it at around 50 μA remains almost unchanged whether the barrier is unmagnetized or
not, consistent with the results of the second lobes. This result can prove that the
in-plane magnetization here has no apparent suppression on the Cooper pair tunneling
in our JJs, instead, it merely serves as the net magnetic flux added to the junction
leading to the observed field modulations.

**Fig. R12. Comparison of the maximum I_c in the Fraunhofer patterns with the**
**barrier at different magnetized states. (a)** The behavior of I_c starting from different
external in-plane magnetic fields. Left (black): Initial magnetic field application
sequence, corresponding to the initially unmagnetized barrier saturated by subjecting
it to higher positive fields. Middle (red): Increasing external magnetic field starting
from -9 mT to positive values. Right (blue): Increasing external magnetic field
starting from -15 mT to positive values. The vertical dashed lines indicate the
positions of their central maximum I_c , while the horizontal dashed lines indicate a
nearly identical maximum I_c under these three magnetizing processes both for the
main and second lobes of the Fraunhofer patterns.

At last, we want to explain our interpretations of the mechanism depicted in **Fig.**
 **4(a)**. The use of this model is to simply explain the birth of 0 and π phase in our CGT
 JJs due to the momentum conserving tunneling between the Cooper pairs at separated
 layers, which is one of the possible scenarios considering the barrier's real magnetic
 structures at around zero magnetic fields. A sufficiently large junction area is
 inevitable to form a multi-domain state as the spins with different orientations are
 located at several segments, nevertheless the required magnetic field to reverse the
 out-of-plane spins is larger than the observed hysteresis in our devices. Instead, the
 change of in-plane domains⁶¹ will presumably lead to the abnormal field modulations
 of the I_c . Under the inhomogeneous magnetism, the saturated state will have a
 significant net moment that is parallel to the applied magnetic field, resulting in the
 early appearance of successive peaks of $I_c(B)$ at lower fields. Also, the heights of the
 subsequent peaks of the $I_c(B)$ or $R(B)$ along with the field widths of the oscillation
 periods at higher fields have deviated from the standard Fraunhofer diffraction
 envelopes, coincident well to the phenomenon that appeared in device #01 as shown
 in **Fig. R13(a)**.

To conclude, based on the results above, we prefer that it is the barrier's in-plane
 magnetization that leads to the observed hysteresis in our JJs rather than the change of
 out-of-plane spins or domains upon sweeping the in-plane magnetic field. By tuning
 the in-plane domains we may expect the homologous variations in the Fraunhofer
 patterns similar to our results, and the detailed analysis of the $I_c(B)$ or the hysteresis
 will generally require direct visualizations of the specific micromagnetic structure.

 **Fig. R13. Abnormal magnetic field modulations in device #01 at higher fields. (a)**
 The behavior of $R(B)$ (log-scale) at two sweeping branches (red for up sweeping from
 negative to positive fields and blue for down sweeping), in which the oscillation
 periods at higher magnetic fields have deviated from the standard Fraunhofer
 diffraction envelopes (guided by the black arrows), presumably lead by the change of

in-plane domains in the CGT barrier.

**Comment 2:**

*In Fig. 4a-b, a minimum is observed at fields shifted away from $B=0$ T. However,*
*this does not need to be a signature of ϕ -JJ. In an asymmetric 0 - π JJ, this feature is*
*commonly observed even if its ground state is 0 or π . An effective method is to follow*
*the standard sweeping scheme shown in many works, e.g. PRL 109, 107002 (2012)*
*and find the four critical current in the I_c - B phase diagram. Considering the*
*importance of ϕ -JJ, such an experiment is strongly suggested.*

**Response:**

We thank the reviewer for the kind suggestions on determining the phase shifts
in our JJs by more effective methods like the applications of standard sweeping
schemes to confirm the switching currents. According to our results, in the section
named “Evidence of π phase coupling”, we intend to attribute the observations of
non-Fraunhofer pattern with a central minimum of critical current (I_c) to the
consequence of a likely 0 - π JJ structure, as we have excluded the influence of
Abrikosov vortex which might also lead to a similar signature in the I_c patterns.
Origin of the π ground phase through the CGT barrier is worthy of discussion, and we
have put forward two possible scenarios to explain: one is from the
momentum-conserving tunneling process of the spins in CGT which generates a
π -shift to the supercurrent^{38,39}, the other is from spin-mixing effects at the magnetic
inhomogeneous S/F interfaces with nonuniform exchange interactions, where the
spin-dependent scattering phases are close to π ⁴⁰⁻⁴⁴. We have to admit that we did not
reveal the actual physical process conclusively in our JJ experiments, and the
existence of the π phase is not so persuasively merely supported by the
non-Fraunhofer patterns. Therefore we have carried out more characterizations to
define the ground state phases in NbSe₂/CGT JJ systems following your suggestions.

It is known that a JJ with an arbitrary value ϕ of the phase between 0 and π
named ϕ -JJ possesses doubly degenerate ground states, which is of great potential
to be utilized as a component for superconducting quantum devices⁴⁷⁻⁵⁰. Its Josephson
energy profile as a double-well potential should give the ground states phases of $\pm\phi$.
Upon applying a bias current to the JJ corresponding to a tilt of the potential, the
phase could be trapped either in the $+\phi$ or $-\phi$ state following a specific bias sweeping
sequence^{51,52}. When switching the JJ from the zero-voltage state to the voltage-state,
the required tilt for the escape of the phase out of the potential well depends on the
initial state ($+\phi$ or $-\phi$) of the junction, at a higher I_{c+} or a lower I_{c-} alternately as we
could get in positive bias currents. Goldobin et al^{48,50,52} have demonstrated the
possibility to use $I_{c\pm}$ for the readout of the $\pm\phi$ state. However, the I_{c-} can only be
observed together with the I_{c+} for low damping^{48,53} (underdamped JJ) allowing a
stationary phase motion, because the probability to find the phase trapped in a certain
state is nearly constant for $\pm\phi$. In contrast, for the regime of high damping
(overdamped JJ), the retrapping process is deterministic to predict the particular
destination well of the Josephson phase, therefore only the higher I_{c+} can be measured.

Due to non-negligible electronic noises in the external circuits of our instruments, as
 we have suggested in the main text, we were not able to evaluate the damping in our
 JJs accurately^{54,55}, since their IVCs show tiny hystereses in the process of switching
 and retrapping like high damping JJs. For device #02 in which the unconventional
 Fraunhofer pattern was observed, we used two schemes to measure the two switching
 currents at zero magnetic field as predicted in a ϕ -JJ. The initial state of the JJ is
 prepared by sweeping from a large positive bias to zero. According to the comparison
 in **Fig. R14(a)**, a switching current I_{c0} got by sweeping from zero to positive bias is
 almost identical to the other one I_{c-} , got by sweeping from large negative bias to
 positive bias. In other words, we failed to observe the evidence of a ϕ -JJ following
 this method because of the high damping circumstance, so we tried another
 phase-sensitive measurement employing a SQUID structure^{45,46} to support our
 assumptions.

 **Fig. R14. Sweeping schemes dependence of switching currents of device #02 at**
 **zero magnetic field. (a)** A switching current I_{c0} is obtained by sweeping from large
 positive bias to zero bias and then back to positive bias (added 1.5 mV offset for
 clarity), while another switching current I_{c-} is obtained by sweeping from large
 negative bias to positive currents. The comparison of these two currents labeled by the
 black dashed line gives the almost identical value, representing the failure in
 determining the ϕ -JJ due to the high damping circumstance.

To detect the phase shift in our systems, we have constructed the SQUIDs
 consisting of an SFS JJ (NbSe₂/CGT) and a reference JJ (NbSe₂/NbSe₂). Details of the
 fabrication process could be read in **Methods**. **Fig. R15(a)** displays the false-color
 SEM image of SQUID #07, and its transport results as SQUID oscillations in critical
 current at 1.85 K are presented in **Fig. R15(b)**. The SQUID critical current as a

function of magnetic flux Φ is counted through the SQUID loop ($17.5 \mu\text{m}^2$) as the
 sum of the empty ($\sim 5.9 \mu\text{m}^2$) areas and screening areas ($\sim 11.6 \mu\text{m}^2$), agreed with the
 area calculated from the oscillation period ($\sim 0.118 \text{ mT}$), as shown in **Fig. R15(c)**. To
 extract the phase of the SQUID #07, we have calibrated the measurement systems
 through a NbSe₂ SQUID on the same chip to give zero magnetic field, the data of
 which is also plotted in **Fig. R15(c)**. Under this condition we can find that the position
 of maximum critical current in one periodicity of SQUID #07 is deviated from zero
 magnetic field (guided by the red line), providing a nontrivial phase $\varphi = 148.6^\circ$ rather
 than a 0 or π phase, corresponding to the higher I_{c+} in our experiments. Although the
 lower I_{c-} for the other phase is not capable to realize in our high damping JJs, the
 nontrivial phase (not 0 or π) of SQUID #07 could still be regarded as evidence to
 support the π phase coupling.

 **Fig. R15. Josephson phase measurements of SQUID #07.** (a) False-color SEM
 image of SQUID #07 (scale bar: $2 \mu\text{m}$), consisting of an SFS JJ with a CGT barrier
 and an intrinsic reference JJ formed by two NbSe₂. The area of the SQUID loop (17.5
 878 μm^2) is the sum of the empty parts ($\sim 5.9 \mu\text{m}^2$) and the screening areas ($\sim 11.6 \mu\text{m}^2$). (b)
 Differential resistance dV/dI (log-scale) map of SQUID #07 as a function of bias
 current I and out-of-plane magnetic field measured at 1.85 K , showing a SQUID
 oscillation. All the dV/dI curves are sweeping from zero to positive bias. The position
 of zero magnetic field calibrated by a NbSe₂ SQUID on the same substrate chip is
 indicated by the purple line in this plot. (c) Critical current I_c oscillations of SQUID
 #07 (black rectangles) with the period of 0.118 mT , consistent with the SQUID loop's
 area. The phase φ of it is about 148.6° , obtained by a fit to I_c with the sine relation
 (black lines). The red circles show the I_c oscillations of the NbSe₂ SQUID for
 calibration with the period of 0.209 mT . (d) Temperature dependence of I_c of SQUID
 #07 at zero magnetic field. The thickness of the CGT barrier is $\sim 3.5 \text{ nm}$ measured by

AFM. The nonmonotonic change of I_c implies an incomplete $0-\pi$ transition in SQUID
#07 with a ϕ ground state phase at low temperature.

To further address the ground state phase, we have measured the critical current
of SQUID #07 in the variation of temperature, and surprisingly found a
non-monotonic dependence which is likely to be an induced $0-\pi$ transition occurring
at $T \sim 0.8 T_c$ in this device, as shown in **Fig. R15(d)**. The thickness of the CGT barrier
is about 3.5 ± 0.7 nm measured by AFM. Compared with metallic ferromagnetic π -JJ
which requires the thickness of the barrier to be comparable to the characteristic
length of the order-parameter's damped oscillation, the decay length of insulator CGT
was measured by **Idzuchi et al**³⁹ to be around 1.4 nm, much thinner than the barriers
we used within 3.5~11 nm. The unexpected long-range correlations existing in such
thick ferromagnetic JJs might imply the presence of spin-triplet supercurrents⁵⁶⁻⁵⁸,
where the magnitude of the critical current decays much slowly with barrier thickness
increasing. This phenomenon has been extensively studied in metallic ferromagnetic
JJs, while **Caruso et al**⁴⁴ first discovered the signatures of spin-polarized triplet pairs
in the insulating-barrier JJs NbN/GdN/NbN. According to their results, nonuniform
exchange interactions, as well as spin-triplet correlations, play an important role to
explain the incomplete $0-\pi$ transition they observed in the temperature dependence of
the critical current $I_c(T)$, different from the complete transition due to the presence of
homogeneous exchange fields inside a metallic ferromagnet. Based on the assumption
of interfacial inhomogeneities with high spin-filter efficiency of the barrier and a large
spin mixing angle between the different transport channels, stronger magnetic activity
is expected to be dominant in longer JJs, in which the deviations from the
conventional Ambegaokar Baratoff (AB) behavior are more evident for thicker barrier
samples as the experimental data. Moreover, their recent investigations^{59,60} showed
how the spin-orbit coupling (SOC) interaction as spin-mixing mechanisms and on-site
magnetic impurities affect this unconventional $0-\pi$ transition, essentially due to the
variation of relative amplitudes of different spin correlation functions. Tuning the
strength of SOC or the impurity potential can stabilize the system into the 0 or the π
energy state over the whole range of temperatures.

When referring to our results, the shape of the $I_c(T)$ curve in SQUID #07 is a
more standard cusp-like $0-\pi$ transition with a higher weight of the s-wave singlet
component, corresponding to the relatively low values of disorder and spin-mixing
effects. Since the origin of a ϕ ground state phase comes from the lateral combination
of 0 -JJ and π -JJ parts coexisting in the same junction region, either of them may
undergo the $0-\pi$ or $\pi-0$ transitions at different temperatures, therein we are not able to
determine the phase at higher temperature to be either at 0 or π . Anyway, we believe
the observations above are convincing enough to support our interpretation of the
authenticity of π phase coupling in our NbSe₂/CGT JJs. Combined with the evidence
addressed previously as the non-Fraunhofer patterns, to clearly explore how the
ground state phase as well as the spin-orbit interaction or impurity potential evolves
with barrier thickness, and also to take into account the realistic micromagnetic nature
of barriers like multi-domain structures, worths further investigations on the regime

adapted for full-range thickness in such a ferromagnetic insulator JJ system.

*By the way, the I_c - B pattern should change significantly as a function of the*
*azimuthal angle of B fields, as long as there are strip domains in CGT. Especially*
*when the field is perpendicular to the domain walls, it should restore to a Fraunhofer*
*pattern. This is useful to exclude many possible scenarios and should be carried out.*

**Response:**

We are extremely grateful for your professional suggestion to reveal the nature
of the CGT barrier through experiments related to the azimuthal angle of magnetic
fields. However, we are so sorry that it is kind of hard for us to carry out such
experiments based on the instruments we have up to now. On the one hand, the probe
station to hold the substrate is not able to rotate as in the commercial sample rod
purchased from Quantum Design at ultracold temperatures (below 1 K). On the other
hand, we do have a rotation rod equipped for the PPMS system (above 1.9 K),
nevertheless, the difficulty in making high performance CGT JJs limits our success
rate to realize a nearly zero residual resistance sample at such a higher temperature.
We have fabricated over 120 samples but only a few of them meet our requirements.
To overcome these two obstacles, we are in progress to install the newly self-made
instruments which allow us to rotate the sample at ultracold temperatures, yet to some
extent, we find it impossible to realize within several months. Hence we hope the
added experiments and data above could answer your questions at this time.

**Comment 3:**

*The Fiske resonance is easier identified in the voltage (rather than current)*
*driven experiment. As an important feature of the observation, is there more*
*characterization in addition to Fig. S2?*

**Response:**

Indeed the phenomenon of Fiske resonance is more essential driven by the
voltage rather than the current as the reviewer advised. According to the Josephson
equations, the supercurrent across a JJ oscillates at a constant frequency $\omega_J = 2\pi V/\Phi_0$
at an applied voltage V^{62} . When the oscillation frequency matches the n th harmonic of
the JJ cavity mode as the spatial period of the Josephson current distribution matches
the spatial period of the n th resonant electromagnetic mode of the junction, achieved
by applying a magnetic field, it will result in the excitation of the cavity modes as the
self-induced current steps at specific voltages V_n denoted as Fiske steps^{63–65}.
Regrettably, we have used the current bias to drive the JJ in our experiments and
mainly focused on its critical current, therefore we did not try the voltage-driven
operations to reproduce the Fiske resonance for a comparison. Nevertheless, we try to
integrate the current into the voltage coordinate by the differential resistance dV/dI to
reveal the complementary details of the observed Fiske steps results.

The comparison between the Fiske steps' configuration driven by the current and
the voltage is presented in **Fig. R16(a)**. We can see that in the lower panel, the shaded

areas indicated by the black and white dashed circles represent the specific excitation
 voltages V_1 and V_2 corresponding to the first and second-order Fiske resonances
 respectively. Although the original current bias acting as the coordinate has relatively
 higher accuracy in determining the peak positions of dV/dI , we can only roughly
 define the resonant modes in its integration plots because of the highly damping
 environment in our JJs as **Fig. R16(b)** showed, where the resonances are not sharp
 enough at the almost constant voltages. Nevertheless, we believe that it is more
 accurate than our previous definition of the first order Fiske resonance voltage V_1
 from the inflection point of its current-voltage (IV) curves as we used in **Fig. S2**. The
 value of V_1 is redefined as the valley position of dV/dI vs V corrected from 0.64 mV
 (**Fig. S2**) to 0.57 mV (**Fig. R17(b)**), and the subsequent specific capacitance C_s and
 the parameter β_c mentioned in the main text should be recalculated as 22.3-26.9
 fF/ μm^2 and 5.4-6.7, still in the same order of magnitude as former estimations, which
 does not affect our conclusions.

**Fig. R16. Fiske resonances in device #01 driven by the voltage.** (a) Comparison of
 signatures of the Fiske resonances in the current (upper panel) and the voltage (lower
 panel) coordinate. The first and the second-order Fiske steps are labeled by the black
 (V_1) and white (V_2) arrows with slight broadening due to the errors of integration. (b)
 dV/dI as a function of current (black) and its integrating voltage (red) measured at
 zero magnetic field.

More characterizations are presented next to confirm the flux dependence of the
 resonant modes, in which we select five line cuts of the voltage-driven plots at the
 different magnetic fields in the case of successive periods of its diffraction pattern,
 indicated as **Fig. R17(a)**. The valley positions of the curves measured at 4, 9.5 and 15
 mT shown in **Fig. R17(c)** are nearly the same regardless of slight shifts, accounting
 for the clear first-order Fiske resonance at around 0.57 mV. The second-order Fiske
 resonance is observed in the curve of 12.5 mT at around 0.97 mV while the first-order
 step disappears. Compared to the single resonant mode results, they simultaneously

occur in the curve of 18.5 mT shown in **Fig. R17(d)**, and the resonance voltages are
 very close to the values above. Therefore we can find it easier to identify the Fiske
 resonances in the voltage-driven plots as the reviewer suggested. To precisely
 determine the resonant voltages, we expect further improvements in the devices'
 qualities for negligible damping circumstance and higher-order Fiske steps, under
 which the internal losses in our JJs can be revealed more accurately.

 **Fig. R17. Flux dependence of the resonant modes driven by the voltage in device**
 **#01.** (a) Fiske resonances in the voltage coordinate. The dashed lines mark out the
 positions of successive quantum fluxes embedded in the junction (yellow: 4 mT for
 Φ_0 ; blue: 9.5 mT for $2\Phi_0$; green: 12.5 mT for $2.5\Phi_0$; pink: 15 mT for $3\Phi_0$; purple:
 18.5 mT for $4\Phi_0$). (b) I (black) along with dV/dI (red) as a function of bias V
 measured at 4 mT in-plane magnetic field. The dip in its dV/dI represents the first
 order Fiske step at $V_1 = 0.57$ mV. (c) dV/dI vs V curves measured at 4, 9.5, and 15 mT,
 where the orders of Φ_0 equal to integers and only the first Fiske resonant mode can be
 seen. (d) dV/dI vs V curves measured at 4, 12.5, and 18.5 mT, where the second Fiske
 resonant mode V_2 is at about 0.97 mV.

 **Comment 4:**

*For devices #2 and #3, the area of JJ should be presented, so as to support the*
 *conclusion that the multi-domain structures results from thickness rather than the*
 *large lateral dimension.*

 **Response:**

The areas of device #02 and #03 are $19.6 \mu\text{m}^2$ and $23.5 \mu\text{m}^2$ respectively as
 labeled in **Fig. R18(a)** and **(b)**, kind of larger than the area of device #01 ($14.2 \mu\text{m}^2$)
 which has no obvious signatures of unconventional Fraunhofer patterns like the other
 two JJs. As previously reported, for multiple domain structures in CGT flakes^{14,23,27,28},
 the characteristic domain size is around 70~120 nm which is much smaller than the
 junction width, therefore we may expect the possible existence of a multi-domain
 state merely considering the lateral sizes of our all JJs. It is thus required to justify the
 role of CGT thickness in the formation of multi-domains.

Combined with our supplementary experiments on SQUID #07 in which a ϕ
 ground state phase is observed with the barrier thickness of around 4 nm and the
 junction area of $6.5 \mu\text{m}^2$ (only for the CGT JJ part), we also find it reasonable to
 assume the coexistence of several regions with different magnetic orientations leading
 to a series of 0 and π phase segments in such a thinner CGT. That is, the multi-domain
 structures are possibly occurring through the whole barrier thickness range of our
 CGT JJs from 4 to 11 nm, and we are not able to attribute the tendency of their
 formations to the thickness in the absence of a detailed understanding of their
 micromagnetics based on our existing proofs.

**Fig. R18. Areas of JJ devices #02 and #03.** (a) Optical image of device #02 (scale
 1053 bar: $5 \mu\text{m}$) with a junction area of about $19.6 \mu\text{m}^2$. (b) Optical image of device #03
 (scale bar: $5 \mu\text{m}$) with a junction area of about $23.5 \mu\text{m}^2$

**Additional Comments:**

*In addition, there are some unclear statements that need to be improved:*

*1.Line 164, 'the two nonequivalent S/F hetero-interfaces vertically'. Why is it*
 *related to the asymmetric values at +/- B fields?*

**Response:**

We are sorry to mistake the asymmetric values of hysteresis at positive and
 negative fields for the reason of two vertically nonequivalent S/F hetero-interfaces.
 The unequal interfaces may lead to a difference in the JJ's critical current initially
 measured between the positive and negative directional bias as the supercurrent flows

across the barrier, yet not expected concerning the directions of in-plane magnetic
 fields. According to the data shown in **Fig. 2(c)**, the asymmetric hystereses identified
 as the separations of the resistance peaks are directly accounting for the slight
 variation of diffraction periods at two sweeping branches. The field widths of the
 central lobe, as well as the first order side lobes, are summarized in **Fig. R19(a)**, from
 which the values got when magnetic field sweeps from negative to positive are about
 0.2 mT larger than those at the other direction. If they are closer, then the hysteresis
 values at positive and negative magnetic fields will be more identical to each other.
 As the difference between the periods may be originated from the asymmetry of the
 barrier's in-plane magnetizations and the magnitude of it is within the errors to
 estimate the hysteresis, we will remove such a controversial statement in the main
 text.

**Fig. R19. Comparison of the periods near zero magnetic field at two sweeping**
 **branches in device #01. (a)** Summary of the field widths of the central and the
 first-order side lobes (blue for down sweeping from positive to negative magnetic
 fields and red for up sweeping), while the latter ones are about 0.2 mT larger than the
 former values. The black arrows indicate the correspondence of the side lobes risen by
 the field-induced distortion.

2.Line 169, 'still' is not necessary; 'at higher fields', not 'field'.

Response:

Great thanks to the reviewer's suggestion. We have revised it in the main text.

3.The derivation of the lower panel of Fig. 3a indicated by line 173 should be detailed
 in SI.

**Response:**

Great thanks to the reviewer's suggestion. We have added the detailed derivation
in Supplementary Materials.

Following the procedure called Josephson magnetometry^{66,67}, the experimental
$I_c(B)$ or $R(B)$ curves can be transformed into the $M(B)$ curves, as the positions of the
maximum (minimum) and minimum (maximum) of the I_c (R) are determined by the
relations

$$1103 \quad I_c(B): \Phi^{(min)} = m\Phi_0, \Phi^{(max)} = \left(n + \frac{1}{2}\right)\Phi_0, \tan\left[\frac{\pi\Phi^{(max)}}{\Phi_0}\right] = \frac{\pi\Phi^{(max)}}{\Phi_0}$$

$$1104 \quad R(B): \Phi^{(min)} = \left(n + \frac{1}{2}\right)\Phi_0, \Phi^{(max)} = m\Phi_0, \tan\left[\frac{\pi\Phi^{(min)}}{\Phi_0}\right] = \frac{\pi\Phi^{(min)}}{\Phi_0}$$

where m and n are integers.

As the magnetic homogeneity of the interlayer is satisfied, the magnetic flux Φ
through the junction is the sum of the flux Φ_M due to the ferromagnetic interlayer's
magnetization M and the external field flux Φ_H

$$\Phi = \Phi_M + \Phi_H = 4\pi MWd_F + \Phi_H$$

where W is the junction width perpendicular to the magnetic field, d_F is the thickness
of the ferromagnetic barrier and Φ_H is proportional to the magnetic field. When we
get the $R(B)$ curves as shown in the upper panel in **Fig. 3(a)**, we first define the
central lobe as the zero-order index ($n = 0$), and the successive peaks and valleys are
determined as the positions of the index equal to $\pm 1, \pm 3/2, \pm 2, \dots$. Using these
relations we can transform $R(B)$ into $\Phi(B)$ dependence as the result presented in the
lower panel of **Fig. 3(a)**.

The $\Phi(B)$ curves can further be transformed into $M(B)$ curves by subtracting the
external field flux term and the results are compared in **Fig. R20(a)**. The magnitudes
of the interlayer's remanences and the saturation magnetizations can be read from the
$M(B)$ curves. However due to the nonuniformity of periods of this device especially at
high fields, rough approximations of them are expected rather than the exact
estimations based on our data, therefore the direct derivations as the $\Phi(B)$ curves are
preferred in the main text.

**Fig. R20. Extracted $\Phi(B)$ and $M(B)$ from $R(B)$ in device #01. (a)** Magnetic field
 dependence of trapped flux in the junction (upper), with each point representing
 integer or half-integer Φ_0 corresponding to the maximum or minimum in $R(B)$. The
 barrier's magnetization (lower) curves $M(B)$ are obtained from the data in $\Phi(B)$.

*4.Line 242, the magnitude of coherence length of NbSe₂ should be presented.*

**Response:**

The magnitudes of the coherence length of NbSe₂ parallel or perpendicular to the
 layers are around 7-11 nm and 2-3 nm as previously reported [refs 1,68–70].

*5.Line 308, 'gained' is confusing.*

**Response:**

We are sorry to use an inappropriate statement here, and we have changed the
 word 'gained' to 'measured'.

*6.Line 535, the magnetic structure should be added to the cartoon of CGT, and the
 pinned spin in NbSe₂.*

**Response:**

Great thanks to the reviewer's suggestion. We have added it to the atomic
 structures of CGT in **Fig. 1(a)** and the illustration of spin-momentum locking in
 NbSe₂ in **Fig. 1(b)**.

**Based on the reviewers' comments, we have performed the following corrections:**

**Revised manuscript:**

- 1. Page 4: We have recalculated the values (including the specific capacitance C_s and
the parameter β_c) by the corrected first order Fiske resonant voltage V_I , obtained
from the dV/dI curves in the coordinate of driven V .
- 2. Page 4: We have deleted the wrong statement ‘Nevertheless, the values measured
at the negative magnetic field are smaller than those at the positive field,
suggesting an asymmetric supercurrent distribution from the two nonequivalent
S/F hetero-interfaces vertically’.
- 3. Page 4: We have corrected the expression here to be ‘Empirically, we can
correspond each minimum/maximum of R to half-integer/integer flux quantum
trapped in the junction’.
- 4. Page 4: We have corrected the equation here to be $\Phi = 4\pi MWd_F + \Phi_H$.
- 5. Page 5 and 8: We have added the areas of device #02 and device #03.
- 6. Page 6: We have added the magnitude of coherence length perpendicular to the
layers of NbSe₂.
- 7. Page 6-7: We have added the descriptions of our new results of SQUID #04 to
jointly support the ‘**Evidence of π phase coupling**’ in our CGT JJs with the
unconventional Fraunhofer patterns.
- 8. Page 7-8: We have rearranged the interpretations of trapped vortex and
asymmetries in ferromagnetic interlayers from ‘**Evidence of π phase coupling**’ to
**Discussion** for a better organization.

**Revised figure and table:**

- 1. Fig. 1(a) and (b): We have revised them as the new versions added with necessary
illustrations.
- 2. Fig. 3(a): We have corrected it to the new version, for we have mistaken the ± 1
order into the $\pm 1/2$ order.
- 3. Fig. 4: We have changed the title of its figure caption for a more explicit
description.
- 4. Fig. 5: We have added the results of Josephson phase measurements to further
support the assumptions on the π phase coupling in our CGT JJs.

**Revised supplementary materials:**

- 1. Supplementary Section I: We have added more characterizations on the Fiske
resonances in addition to Fig. S2 in the coordinate of voltage.
- 2. Supplementary Section VII: We have added results of the critical currents got in
device #02 in different sweeping schemes to show the influences of the high
damping environment on our JJs.
- 3. Supplementary Section VIII: We have added the detailed derivation of the lower
panel of Fig. 3(a) as the extracted $\Phi(B)$ results from the original $R(B)$ curves.

**Overall, we thank all the Reviewers for their detailed comments and**
**valuable suggestions. We have made the necessary revisions based on these**
**comments and hope that our response is satisfactory to the reviewers. We also**

appreciate the significant effort that the reviewers have dedicated to this
manuscript and the dramatic help to further improve our manuscript. Thank
you all.

Reference

- 1. de Trey, P. & Gygax, S. Anisotropy of the Ginzburg-Landau Parameter κ in NbSe₂. *J*
*Low Temp Phys* **11**, 421–434 (1973).
- 2. Le, L. P. *et al.* Magnetic penetration depth in layered compound NbSe₂ measured
by muon spin relaxation. *Phys. C* **185–189**, 2715–2716 (1991).
- 3. Fletcher, J. D. *et al.* Penetration Depth Study of Superconducting Gap Structure of
2H – NbSe₂. *Phys. Rev. Lett.* **98**, 057003 (2007).
- 4. Barone, A. & Paternò, G. *Physics and applications of the Josephson effect.* (Wiley,
1982).
- 5. Suominen, H. *et al.* Anomalous Fraunhofer interference in epitaxial
superconductor-semiconductor Josephson junctions. *Phys. Rev. B* **95**, 035307
(2017).
- 6. Barati, F. *et al.* Tuning Supercurrent in Josephson Field-Effect Transistors Using
h-BN Dielectric. *Nano Lett.* **21**, 1915–1920 (2021).
- 7. Wang, F., Macià, F., Wohlgenannt, M., Kent, A. D. & Flatté, M. E. Magnetic
Fringe-Field Control of Electronic Transport in an Organic Film. *Phys. Rev. X* **2**,
021013 (2012).
- 8. Salis, G., Alvarado, S., Tschudy, M., Brunswiler, T. & Allenspach, R. Hysteretic
electroluminescence in organic light-emitting diodes for spin injection. *Phys. Rev. B*
**70**, 085203 (2004).
- 9. Marton, L., Lachenbruch, S. H., Simpson, J. A. & VanBronkhorst, A. Fringe Fields of
Ferromagnetic Domains. *J. Appl. Phys.* **20**, 1258–1258 (1949).
- 10. Thomas, L., Samant, M. G. & Parkin, S. S. P. Domain-Wall Induced Coupling
between Ferromagnetic Layers. *Phys. Rev. Lett.* **84**, 1816–1819 (2000).
- 11. You, C.-Y. & Bader, S. D. Enhancement of switching stability of tunneling
magnetoresistance systems with artificial ferrimagnets. *J Appl Phys* **92**, 3886–3889
(2002).
- 12. You, C.-Y. Confined magnetic stray field from a narrow domain wall. *J Appl Phys*
**100**, 043911 (1–4) (2006).
- 13. A. H. Aharoni. Introduction to the Theory of Ferromagnetism. (1996).
- 14. Han, M.-G. *et al.* Topological Magnetic-Spin Textures in Two-Dimensional van der
Waals Cr₂Ge₂Te₆. *Nano Lett.* **19**, 7859–7865 (2019).
- 15. Gong, C. *et al.* Discovery of intrinsic ferromagnetism in two-dimensional van der
Waals crystals. *Nature* **546**, 265–269 (2017).
- 16. Zhang, X. *et al.* Magnetic anisotropy of the single-crystalline ferromagnetic
insulator Cr₂Ge₂Te₆. *Jpn. J. Appl. Phys.* **55**, 033001 (2016).
- 17. Lin, Z. *et al.* Pressure-induced spin reorientation transition in layered

- ferromagnetic insulator $\text{Cr}_2\text{Ge}_2\text{Te}_6$. *Phys. Rev. Mater.* **2**, 051004 (2018).
- 18. Liu, W. *et al.* Anisotropic magnetoresistance behaviors in the layered
ferromagnetic $\text{Cr}_2\text{Ge}_2\text{Te}_6$. *J. Phys. Appl. Phys.* **53**, 025101 (2020).
- 19. Mogi, M. *et al.* Ferromagnetic insulator $\text{Cr}_2\text{Ge}_2\text{Te}_6$ thin films with perpendicular
remanence. *APL Mater.* **6**, 091104 (2018).
- 20. Selter, S., Bastien, G., Wolter, A. U. B., Aswartham, S. & Büchner, B. Magnetic
anisotropy and low-field magnetic phase diagram of the quasi-two-dimensional
ferromagnet $\text{Cr}_2\text{Ge}_2\text{Te}_6$. *Phys. Rev. B* **101**, 014440 (1–10) (2020).
- 21. Zhang, T. *et al.* Laser-induced magnetization dynamics in a van der Waals
ferromagnetic $\text{Cr}_2\text{Ge}_2\text{Te}_6$ nanoflake. *Appl. Phys. Lett.* **116**, 223103 (2020).
- 22. Fang, Y., Wu, S., Zhu, Z.-Z. & Guo, G.-Y. Large magneto-optical effects and
magnetic anisotropy energy in two-dimensional $\text{Cr}_2\text{Ge}_2\text{Te}_6$. *Phys. Rev. B* **98**, 125416
(2018).
- 23. Lohmann, M. *et al.* Probing Magnetism in Insulating $\text{Cr}_2\text{Ge}_2\text{Te}_6$ by Induced
Anomalous Hall Effect in Pt. *Nano Lett.* **19**, 2397–2403 (2019).
- 24. Kooy, C. Experimental and theoretical study of the domain configuration in thin
layers of $\text{BaFe}_{12}\text{O}_{19}$. *Philips Res Repts* **15**, 7 (1960).
- 25. Kaplan, B. & Gehring, G. A. The domain structure in ultrathin magnetic films. *J.*
*Magn. Magn. Mater.* **128**, 111–116 (1993).
- 26. Kim, K.-J. & Choe, S.-B. Analytic theory of wall configuration and depinning
mechanism in magnetic nanostructure with perpendicular magnetic anisotropy. *J.*
*Magn. Magn. Mater.* **321**, 2197–2199 (2009).
- 27. Wang, Z. *et al.* Electric-field control of magnetism in a few-layered van der Waals
ferromagnetic semiconductor. *Nat. Nanotechnol.* **13**, 554–559 (2018).
- 28. Guo, T. *et al.* Multiple domain structure and symmetry types in narrow
temperature and magnetic field ranges in layered $\text{Cr}_2\text{Ge}_2\text{Te}_6$ crystal measured by
magnetic force microscope. *Mater. Charact.* **173**, 110913 (2021).
- 29. Li, Y. F. *et al.* Electronic structure of ferromagnetic semiconductor CrGeTe_3 by
angle-resolved photoemission spectroscopy. *Phys. Rev. B* **98**, 125127(1–6) (2018).
- 30. Senapati, K., Blamire, M. G. & Barber, Z. H. Spin-filter Josephson junctions. *Nat.*
*Mater.* **10**, 849–852 (2011).
- 31. Pal, A., Barber, Z. H., Robinson, J. W. A. & Blamire, M. G. Pure second harmonic
current-phase relation in spin-filter Josephson junctions. *Nat. Commun.* **5**, 3340
(2014).
- 32. Massarotti, D. *et al.* Macroscopic quantum tunnelling in spin filter ferromagnetic
Josephson junctions. *Nat. Commun.* **6**, 7376 (2015).
- 33. Schwidtal, K. & Finnegan, R. D. Barrier - Thickness Dependence of the dc
Quantum Interference Effect in Thin - Film Lead Josephson Junctions. *J. Appl. Phys.*
**40**, 2123–2127 (1969).
- 34. Monaco, R., Koshelets, V. P., Mukhortova, A. & Mygind, J. Self-field effects in
window-type Josephson tunnel junctions. *Supercond. Sci. Technol.* **26**, 055021
(2013).

- 35. Richter, N. *et al.* Temperature-dependent magnetic anisotropy in the layered
magnetic semiconductors CrI₃ and CrBr₃. *Phys. Rev. Mater.* **2**, 024004 (2018).
- 36. Webster, L. & Yan, J.-A. Strain-tunable magnetic anisotropy in monolayer CrCl₃,
CrBr₃, and CrI₃. *Phys. Rev. B* **98**, 144411 (2018).
- 37. Yu, X. *et al.* Large magnetocaloric effect in van der Waals crystal CrBr₃. *Front.*
*Phys.* **14**, 43501 (2019).
- 38. van Dam, J. A., Nazarov, Y. V., Bakkers, E. P. A. M., De Franceschi, S. &
Kouwenhoven, L. P. Supercurrent reversal in quantum dots. *Nature* **442**, 667–670
(2006).
- 39. Idzuchi, H. *et al.* Unconventional supercurrent phase in Ising superconductor
Josephson junction with atomically thin magnetic insulator. *Nat. Commun.* **12**,
5332 (2021).
- 40. Fogelström, M. Josephson currents through spin-active interfaces. *Phys. Rev. B*
**62**, 11812–11819 (2000).
- 41. Holmqvist, C., Teber, S. & Fogelström, M. Nonequilibrium effects in a Josephson
junction coupled to a precessing spin. *Phys. Rev. B* **83**, 104521 (2011).
- 42. Kulagina, I. & Linder, J. Spin supercurrent, magnetization dynamics, and ϕ -state
in spin-textured Josephson junctions. *Phys. Rev. B* **90**, 054504 (2014).
- 43. Yokoyama, T., Eto, M. & Nazarov, Y. V. Anomalous Josephson effect induced by
spin-orbit interaction and Zeeman effect in semiconductor nanowires. *Phys. Rev. B*
**89**, 195407 (2014).
- 44. Caruso, R. *et al.* Tuning of Magnetic Activity in Spin-Filter Josephson Junctions
Towards Spin-Triplet Transport. *Phys. Rev. Lett.* **122**, 047002 (2019).
- 45. Guichard, W. *et al.* Phase Sensitive Experiments in Ferromagnetic-Based
Josephson Junctions. *Phys. Rev. Lett.* **90**, 167001 (2003).
- 46. Feofanov, A. K. *et al.* Implementation of superconductor/ferromagnet/
superconductor π -shifters in superconducting digital and quantum circuits. *Nat.*
*Phys.* **6**, 593–597 (2010).
- 47. Mints, R. G. Self-generated flux in Josephson junctions with alternating critical
current density. *Phys. Rev. B* **57**, R3221–R3224 (1998).
- 48. Goldobin, E., Koelle, D., Kleiner, R. & Buzdin, A. Josephson junctions with second
harmonic in the current-phase relation: Properties of ϕ junctions. *Phys. Rev. B* **76**,
224523 (2007).
- 49. Pugach, N. G., Goldobin, E., Kleiner, R. & Koelle, D. Method for reliable realization
of a ϕ Josephson junction. *Phys. Rev. B* **81**, 104513 (2010).
- 50. Goldobin, E., Koelle, D., Kleiner, R. & Mints, R. G. Josephson Junction with a
Magnetic-Field Tunable Ground State. *Phys. Rev. Lett.* **107**, 227001 (2011).
- 51. Sickinger, H. *et al.* Experimental Evidence of a ϕ Josephson Junction. *Phys. Rev.*
*Lett.* **109**, 107002 (2012).
- 52. Goldobin, E. *et al.* Memory cell based on a ϕ Josephson junction. *Appl. Phys. Lett.*
**102**, 242602 (2013).
- 53. Menditto, R. *et al.* Phase retrapping in a ϕ Josephson junction: Onset of the

- butterfly effect. *Phys. Rev. B* **93**, 174506 (2016).
- 54. Martinis, J. M. & Kautz, R. L. Classical phase diffusion in small hysteretic
Josephson junctions. *Phys. Rev. Lett.* **63**, 1507–1510 (1989).
- 55. Kautz, R. L. & Martinis, J. M. Noise-affected I-V curves in small hysteretic
Josephson junctions. **42**, 9903–9938 (1990).
- 56. Khaire, T. S., Khasawneh, M. A., Pratt, W. P. & Birge, N. O. Observation of
Spin-Triplet Superconductivity in Co-Based Josephson Junctions. *Phys. Rev. Lett.*
**104**, 137002 (2010).
- 57. Robinson, J. W. A., Witt, J. D. S. & Blamire, M. G. Controlled Injection of
Spin-Triplet Supercurrents into a Strong Ferromagnet. *Science* **329**, 59–61 (2010).
- 58. Banerjee, N., Robinson, J. W. A. & Blamire, M. G. Reversible control of
spin-polarized supercurrents in ferromagnetic Josephson junctions. *Nat. Commun.*
**5**, 4771 (2014).
- 59. Ahmad, H. G. *et al.* Coexistence and tuning of spin-singlet and triplet transport in
spin-filter Josephson junctions. *ArXiv:2106.15646* (2021).
- 60. Minutillo, M., Capecehatro, R. & Lucignano, P. Realization of $0 - \pi$ states in SFIS
Josephson junctions. The role of spin-orbit interaction and lattice impurities.
*ArXiv:2108.04292* (2021).
- 61. Blamire, M. G., Smiet, C. B., Banerjee, N. & Robinson, J. W. A. Field modulation of
the critical current in magnetic Josephson junctions. *Supercond. Sci. Technol.* **26**,
055017 (2013).
- 62. Chang, J.-J. Fiske steps in Josephson junctions. *Phys. Rev. B* **28**, 1276–1280
(1983).
- 63. Pfeiffer, J. *et al.* Static and dynamic properties of 0 , π , and $0 - \pi$ ferromagnetic
Josephson tunnel junctions. *Phys. Rev. B* **77**, 214507 (2008).
- 64. Petković, I., Aprili, M., Barnes, S. E., Beuneu, F. & Maekawa, S. Direct dynamical
coupling of spin modes and singlet Josephson supercurrent in ferromagnetic
Josephson junctions. *Phys. Rev. B* **80**, 220502 (2009).
- 65. Wild, G., Probst, C., Marx, A. & Gross, R. Josephson coupling and Fiske dynamics
in ferromagnetic tunnel junctions. *Eur. Phys. J. B* **78**, 509–523 (2010).
- 66. Bol'ginov, V. V., Stolyarov, V. S., Sobanin, D. S., Karpovich, A. L. & Ryazanov, V. V.
Magnetic switches based on Nb-PdFe-Nb Josephson junctions with a magnetically
soft ferromagnetic interlayer. *JETP Lett.* **95**, 366–371 (2012).
- 67. Iovan, A., Golod, T. & Krasnov, V. M. Controllable generation of a spin-triplet
supercurrent in a Josephson spin valve. *Phys. Rev. B* **90**, 134514 (2014).
- 68. Banerjee, S. S. *et al.* Magnetic phase diagram of anisotropic superconductor
$2H-NbSe_2$. *Phys. B Condens. Matter* **237–238**, 315–317 (1997).
- 69. El-Bana, M. S. *et al.* Superconductivity in two-dimensional $NbSe_2$ field effect
transistors. *Supercond. Sci. Technol.* **26**, 125020 (2013).
- 70. Nader, A. & Monceau, P. Critical field of $2H-NbSe_2$ down to 50mK. *SpringerPlus* **3**,
16 (2014).

REVIEWERS' COMMENTS

Reviewer #1 (Remarks to the Author):

I appreciate the authors for providing detailed information for the response to the reviewer's comments. I believe that the manuscript is significantly improved after revision and the author's conclusion is strengthened by the additional data sets. I think the results presented in this manuscript are novel and suitable for publication.

Reviewer #2 (Remarks to the Author):

The authors has addressed most of the concerns in the last referee report. In particular, the phase sensitive SQUID measurement and its temperature dependence are helpful to clarify the nontrivial nature of the studied JJ, though the ϕ found is not equal to π . With optimization of device design and fabrication, a ϕ -JJ may be firmly established, which certainly deserves more systematic and in-depth research in the future.

As such, I suggest its publication in nature communications.

Response to Reviewer's Comments

Response to Reviewer #1

General comment:

I appreciate the authors for providing detailed information for the response to the reviewer's comments. I believe that the manuscript is significantly improved after revision and the author's conclusion is strengthened by the additional data sets. I think the results presented in this manuscript are novel and suitable for publication.

Response:

We sincerely thank the reviewer for the recommendation on our responses and our paper for publication.

Response to Reviewer #2

General comment:

The authors has addressed most of the concerns in the last referee report. In particular, the phase sensitive SQUID measurement and its temperature dependence are helpful to clarify the nontrivial nature of the studied JJ, though the ϕ found is not equal to π . With optimization of device design and fabrication, a ϕ -JJ may be firmly established, which certainly deserves more systematic and in-depth research in the future.

Response:

We sincerely thank the reviewer for the recommendation on our responses and our paper for publication.